# LATENT STOCHASTIC INTERPOLANTS

**Saurabh Singh** [*]
Poetiq AI
saurabh@poetiq.ai

**Dmitry Lagun**
Google DeepMind
dlagun@google.com

## ABSTRACT

Stochastic Interpolants (SI) is a powerful framework for generative modeling, capable of flexibly transforming between two probability distributions. However, its use in jointly optimized latent variable models remains unexplored as it requires direct access to the samples from the two distributions. This work presents Latent Stochastic Interpolants (LSI) enabling joint learning in a latent space with end-to-end optimized encoder, decoder and latent SI models. We achieve this by developing a principled Evidence Lower Bound (ELBO) objective derived directly in continuous time. The joint optimization allows LSI to learn effective latent representations along with a generative process that transforms an arbitrary prior distribution into the encoder-defined aggregated posterior. LSI sidesteps the simple priors of the normal diffusion models and mitigates the computational demands of applying SI directly in high-dimensional observation spaces, while preserving the generative flexibility of the SI framework. We demonstrate the efficacy of LSI through comprehensive experiments on the standard large scale ImageNet generation benchmark.

## 1 INTRODUCTION

Diffusion models have achieved remarkable success in modeling complex, high-dimensional data distributions across various domains. These models learn to transform a simple "prior" distribution $p_0$, such as a standard Gaussian, into a complex data distribution $p_1$. While early formulations were constrained to use specific prior distributions that are Lévy Stable, recent advancements, particularly Stochastic Interpolants (SI) (Albergo et al., 2023) offer a powerful, unifying framework capable of bridging arbitrary probability distributions. However, SI assumes that both the prior $p_0$ and the target $p_1$ distributions are fixed and the samples from both are directly *observed*. This requirement limits their use in jointly learned latent variable models where the generative model is learned, along with an encoder and a decoder, in a latent unobserved space. Further, the latent space, often lower dimensional, evolves as the encoder and decoder are jointly optimized. Lack of support for joint optimization implies that arbitrary fixed latent representations may not be optimally aligned with the generative process resulting in inefficiencies.

To address this, we present Latent Stochastic Interpolants (LSI), a novel framework for end-to-end learning of a generative model in an *unobserved* latent space. Our key innovation lies in deriving a principled, flexible and scalable training objective as an Evidence Lower Bound (ELBO) directly in continuous time. This objective, like SI, provides data log-likelihood control, while enabling scalable end-to-end training of the three components: an encoder mapping high-dimensional observations to a latent space, a decoder reconstructing observations from latent representations, and a latent SI model operating entirely within the learned latent space. Our approach allows transforming arbitrary prior distributions into the encoder-defined aggregated posterior, simultaneously aligning data representations with a high-fidelity generative process using that representation.

LSI's single ELBO objective provides a unified, scalable framework that avoids the need for simple priors of the normal diffusion models, mitigates the computational demands of applying SI directly in high-dimensional observation spaces and offers an alternative to ad-hoc multi-stage training. Our formulation admits simulation-free training analogous to observation-space diffusion and SI models, while preserving the flexibility of SI framework. We empirically validate LSI's strengths through

---

[*]Work done while at DeepMind.

comprehensive experiments on the challenging ImageNet generation benchmark, demonstrating competitive generative performance and highlighting its advantages in efficiency.

Our key contributions are: 1) **Latent stochastic interpolants (LSI):** a novel and flexible framework for scalable training of a latent variable generative model with continuous time dynamic latent variables, where the encoder, decoder and latent generative model are jointly trained, 2) **Unifying perspective:** a novel perspective on integrating flexible continuous-time formulation of SI within latent variable models, leveraging insights from continuous time stochastic processes, 3) **Principled ELBO objective:** a new ELBO as a principled training objective that retains strengths of SI – simple simulation free training and flexible prior choice – while enabling the benefits of joint training in a latent space.

## 2 BACKGROUND

**Notation.** We use small letters $x, y, t$ etc. to represent scalar and vector variables, $f, g$ etc. to represent functions, Greek letters $\beta, \theta$ etc. to represent (hyper-)parameters. Lower case letters $x$ are used to represent both the random variable and a particular value $x \sim p(x)$. Dependence on an argument $t$ is indicated as a subscript $u_t$ or argument $u(t)$ interchangeably.

Our work builds upon two key results briefly reviewed below. The first result (Li et al., 2020; Theodorou, 2015) states an Evidence Lower Bound (ELBO) for models using continuous time dynamic latent variables. We state a more general form than the original to aid the discussion of the prior distributions. The second result is a well known method for constructing a stochastic mapping between two distributions. We exploit it to construct a variational approximation in the latent space.

### 2.1 VARIATIONAL LOWER BOUND USING DYNAMIC LATENT VARIABLES

Consider two SDEs, starting with the same starting point $\tilde{z}_0 = z_0$ at $t = 0$, sharing the same dispersion coefficient $\sigma(z_t, t)$ but potentially different initial distributions — $z_0 \sim p_0(z_0)$ for the model, with path measure $\mathbb{P}_\theta$, and $z_0 \sim q_0(z_0)$ for the variational posterior, with path measure $\mathbb{Q}$:

$$d\tilde{z}_t = h_\theta(\tilde{z}_t, t)dt + \sigma(\tilde{z}_t, t)d\tilde{w}_t, \qquad \text{(model, path measure } \mathbb{P}_\theta) \qquad (1)$$

$$dz_t = h_\phi(z_t, t)dt + \sigma(z_t, t)dw_t, \qquad \text{(variational posterior, path measure } \mathbb{Q}) \qquad (2)$$

Where $\tilde{w}_t$ and $w_t$ are Wiener processes under corresponding path measures. The first equation can be viewed as the latent dynamics under the model $h_\theta$ we are interested in learning and the second as the latent dynamics under some variational approximation to the posterior that can be used to produce samples $z_t$. Further, let $x_{t_i}$ be observations at time $t_i$ that are assumed to only depend on the corresponding unobserved latent state $z_{t_i}$, then the ELBO can be written as

$$\ln p_\theta(x_{t_1}, \ldots, x_{t_n}) \geq \mathbb{E}_\mathbb{Q}\left[\sum_{i=1}^n \ln p_\theta(x_{t_i}|z_{t_i}) - \ln \frac{q_0(z_0)}{p_0(z_0)} - \frac{1}{2}\int_0^T \|u(z_t, t)\|^2 \, dt\right] \qquad (3)$$

$$= \mathbb{E}_\mathbb{Q}\left[\sum_{i=1}^n \ln p_\theta(x_{t_i}|z_{t_i})\right] - \text{KL}(\mathbb{Q}\|\mathbb{P}_\theta) \qquad (4)$$

Where $u$ satisfies

$$\sigma(z, t)u(z, t) = h_\phi(z, t) - h_\theta(z, t) \qquad (5)$$

We provide the proof of the above general form in Section A. Similar to the ELBO for the VAEs (Kingma et al., 2013), the first term in eq. (4) explains observations given the latent path and the second term penalizes the mismatch between the variational and model path distributions. In the following, we focus on the case of $q_0 = p_0$ and draw attention to the general case when needed.

### 2.2 DIFFUSION BRIDGE

Given two arbitrary points $z_0$ and $z_1$, a diffusion bridge between the two is a random process constrained to start and end at the two given end points. A diffusion bridge can be used to specify the stochastic dynamics of a particle that starts at $z_0$ at $t = 0$ and is constrained to land at $z_1$ at $t = 1$.

Consider a stochastic process starting at $z_0$ with the dynamics specified by eq. (2). Using Doob's h-transform, the SDE for the end point conditioned diffusion bridge, constrained to end at $z_1$ at time $t = 1$ can be written as

$$dz_t = [h_\phi(z_t, t) + \sigma(z_t, t)\sigma(z_t, t)^T \nabla_{z_t} \ln p(z_1|z_t)]dt + \sigma(z_t, t)dw_t \qquad (6)$$

where $p(z_1|z_t)$ is the conditional density for $z_1$ under the original dynamics in eq. (2) and depends on $h_\phi$. Note that a Brownian bridge is a special case of a Diffusion bridge where the dynamics are specified by the standard Brownian motion. Diffusion bridges can be used to construct a stochastic mapping between two distributions by considering the end points $z_0 \sim p_0(z_0)$ and $z_1 \sim p_1(z_1)$ to be sampled from the two distributions of interest.

## 3 LATENT STOCHASTIC INTERPOLANTS

**Stochastic Interpolants (SI) and their limitation:** SI (Albergo et al., 2023) is a powerful framework for generative modeling, capable of learning a model that can flexibly transform between two probability distributions. Let $x_1 \sim p(x_1)$ be an observation from the data distribution $p(x_1)$ that we want to sample from. In SI framework, another distribution $p_0(x_0)$ is chosen as a prior with samples $x_0 \sim p_0(x_0)$. Typically, $p_0$ is easy to sample from, e.g. a Gaussian distribution. A stochastic interpolant $x_t$ is then constructed with the requirement that the marginal distribution $p_t(x_t)$ of $x_t$ equals $p_0$ at $t = 0$ and $p_1$ at $t = 1$. For example, the interpolant $x_t = (1-t)x_0 + tx_1 + \sqrt{t(1-t)}\epsilon, \epsilon \sim N(0, I)$ satisfies this requirement. The velocity field and the score function for the generative model are then estimated as solutions to particular least squares problems. The learned velocity field and the score function can then be used to transform a sample from $p_0$ to produce a sample from $p_1$. SI requires that the samples $x_0$ and $x_1$ are observed, though $x_1$ could be an output of a *fixed* model, hence still observed. We use the term observation space SI to emphasize this.

However, we are interested in jointly learning a generative model in a latent space to leverage efficiency of low dimensional representations while also aligning the latents with the generative process. Therefore, we want to jointly optimize an encoder $p_\theta(z_1|x_1)$ that represents high dimensional observations in the latent space and a decoder $p_\theta(x_1|z_1)$ that maps a given latent representation to the observation space, along with the generative model in latent space. To use SI, we need to interpolate between a fixed prior $p_0(z_0)$ in the latent space and the true marginal posterior $p_1(z_1) \equiv \int p(z_1|x_1)dx_1$. However, we only have access to the posterior model $p_\theta(z_1|x_1)$ that is optimized concurrently and is an approximation to the true intractable posterior. Consequently, we can not directly construct an interpolant in the latent space that satisfies the requirements of SI. In the following, we address this issue by deriving Latent Stochastic Interpolants (LSI), though from an entirely different perspective than is considered by SI.

**Generative model with dynamic latent variables:** Since we want to jointly learn the generative model in a latent space, we propose a latent variable model where the unobserved latent variables are assumed to evolve in continuous time according to the dynamics specified by an SDE of the form in eq. (1). Let $p_\theta(x_1|z_1)$ be a parameterized stochastic decoder and $h_\theta$ parameterized drift for eq. (1). Then, the generation process using our model is as following – first a sample $z_0 \sim p_0(z_0)$ is produced from a prior $p_0(z_0)$, then $z_0$ evolves according to the dynamics specified by eq. (1) using $h_\theta$ from $t = 0$ to $t = 1$ to yield a $z_1$, and finally an observation space sample is produced using the decoder $p_\theta(x_1|z_1)$. In theory, we can now utilize the ELBO presented in section 2.1 to train this model. Note that, although the ELBO in eq. (3) supports arbitrary number of observations $x_{t_i}$ at arbitrary times $t_i$, in this paper we focus on a single observation $x_1$ at $t = 1$. The ELBO in eq. (3) needs a variational approximation to the posterior $p_\theta(z_t|x_1)$ which can be used to sample $z_t$. This approximation is constructed as another dynamical model specified by the SDE in eq. (2). Unfortunately, for a general variational approximation specified by an arbitrary $h_\phi$, simulating eq. (2) would lead to significant computational burden for large problems during each training iteration and open the door to additional issues resulting from approximations needed for simulation of the SDE. Instead, we explicitly construct the drift $h_\phi$ in eq. (2) such that $z_t$ can be sampled directly without simulation for any time $t$. Our scheme provides a scalable alternative that allows simulation free efficient training, as is common in the observation space diffusion models.

**Variational posterior with simulation free samples:** Next we construct a variational posterior approximation, that enables easy sampling of $z_t$ without requiring the simulation of the SDE in

eq. (2). Let $z_1 \sim p_\theta(z_1|x_1)$ be a stochastic encoding of the observation $x_1$ providing direct access to $z_1$ at $t = 1$. Next, using the Diffusion Bridge specified by eq. (6) we construct a stochastic mapping between the prior $p_0(z_0)$ and the aggregated approximate posterior $\int p_\theta(z_1|x_1)dx_1$ at $t = 1$. The diffusion bridge, coupled with the encoder $p_\theta(z_1|x_1)$ yields our approximate posterior $p_\theta(z_t|x_1)$. However, $p(z_1|z_t)$ is unknown in general. If we additionally assume that $h_\phi(z_t, t) \equiv h_t z_t$ and $\sigma(z_t, t) \equiv \sigma_t$, then the original SDE in eq. (2) becomes linear with additive noise

$$dz_t = h_t z_t dt + \sigma_t dw_t \tag{7}$$

It is well known that for linear SDEs of the above form, the transition density $p(z_t|z_s), t > s$ is gaussian $N(z_t; a_{st}z_s, b_{st}I)$ (see section G) for some functions $a_{st}, b_{st}$ that depend on $h_t, \sigma_t$. Consequently, we can compute $\nabla_{z_t} \ln p(z_1|z_t)$ for a given $z_t$ as

$$\nabla_{z_t} \ln p(z_1|z_t) = \frac{a_{t1}(z_1 - a_{t1}z_t)}{b_{t1}} \tag{8}$$

The transformed SDE in terms of the simplified drift and dispersion coefficients can be expressed as

$$dz_t = [h_t z_t + \sigma_t^2 \nabla_{z_t} \ln p(z_1|z_t)]dt + \sigma_t dw_t \tag{9}$$

Further, if we condition on the starting point $z_0$, then the conditional density $p(z_t|z_1, z_0)$ can be expressed as following using the Bayes rule

$$p(z_t|z_1, z_0) = \frac{p(z_1|z_t, z_0)p(z_t|z_0)}{p(z_1|z_0)} = \frac{p(z_1|z_t)p(z_t|z_0)}{p(z_1|z_0)} \tag{10}$$

where $p(z_1|z_t, z_0) = p(z_1|z_t)$ because of the Markov independence assumption inherent in eq. (2). Note that all the factors on the right are gaussian. It can be shown that the conditional density $p(z_t|z_1, z_0)$ is also gaussian if the transition densities are gaussian and takes the following form

$$p(z_t|z_1, z_0) = \left(\frac{1}{2\pi}\frac{b_{01}}{b_{0t}b_{t1}}\right)^{\frac{d}{2}} \exp\left(-\frac{1}{2}\frac{b_{01}}{b_{0t}b_{t1}}\left\|z_t - \frac{b_{0t}a_{t1}z_1 + b_{t1}a_{0t}z_0}{b_{01}}\right\|^2\right) \tag{11}$$

Where $a_{(\cdot)}, b_{(\cdot)}$ are constant or time dependent scalars and $d$ is the dimensionality of $z_t$. Their specific forms depends on the choice of $h_t, \sigma_t$. Refer to section G for details. $z_t$ can now be directly sampled without simulating the SDE, given a sample $z_0$ and the encoded observation $z_1$. Note that the assumptions made for eq. (7), while restrictive, do not limit the empirical performance.

**Latent stochastic interpolants:** We can now define latent stochastic interpolants using reparameterization trick in conjuction with eq. (11) to parameterize $z_t$ as

$$z_t = \eta_t \epsilon + \kappa_t z_1 + \nu_t z_0, \quad \epsilon \sim N(0, I) \tag{12}$$

For some functions $\eta_t, \kappa_t, \nu_t$ that depend on $a_{(\cdot)}, b_{(\cdot)}$. Note that $\eta_0 = \eta_1 = 0, \kappa_0 = \nu_1 = 0, \kappa_1 = \nu_0 = 1$ since $z_t$ is sampled from a diffusion bridge with the two end points fixed at $z_0, z_1$. Equation (12) specifies a general stochastic interpolant, akin to the proposal in (Albergo et al., 2023), but now in the latent space. If we choose the encoder and decoder to be identity functions, then above can be viewed as an alternative way to construct stochastic interpolants in the observation space. Instead of choosing $h_t, \sigma_t$ first, we can instead choose $\kappa_t, \nu_t$ and infer the corresponding $h_t, \sigma_t$. For example, choosing $\kappa_t = t, \nu_t = 1 - t$ leads to $\sigma_t = \sigma$, a constant, and we arrive at the following

$$z_t = \sigma\sqrt{t(1-t)}\epsilon + tz_1 + (1-t)z_0, \quad \epsilon \sim N(0, I) \tag{13}$$

See section J for a detailed derivation. We use the above form for all the experiments in the paper. Further, if $p_0(z_0)$ is chosen to be a standard gaussian then the interpolant simplifies to $z_t = tz_1 + \sqrt{(1-t)(\sigma^2 t + 1 - t)}z_0$ (section M). With the above interpolants, we can now define the ELBO and optimize it efficiently with simulation free samples $z_t$. We also derive the expressions for variance preserving choices of $\kappa_t = \sqrt{t}, \eta_t^2 + \nu_t^2 = 1 - t$ in section K, however we do not explore this interpolant empirically.

**Constructing training objective using ELBO (eq. (3)):** We first define $u(z_t, t)$ using eq. (9) as

$$u(z_t, t) = \sigma_t^{-1}[h_t z_t + \sigma_t^2 \nabla_{z_t} \ln p(z_1|z_t) - h_\theta(z_t, t)] \quad (14)$$

For the general latent stochastic interpolant $z_t = \eta_t \epsilon + \kappa_t z_1 + \nu_t z_0$ (eq. (12)), we show that $u(z_t, t)$ takes the following form

$$u(z_t, t) = \sigma_t^{-1} \left[ \left( \frac{d\eta_t}{dt} - \frac{\sigma_t^2}{2\eta_t} \right) \epsilon + \frac{d\kappa_t}{dt} z_1 + \frac{d\nu_t}{dt} z_0 - h_\theta(z_t, t) \right] \quad (15)$$

See section H for the proof. This $u(z_t, t)$ can be substituted into the ELBO in eq. (3) to construct a training objective. For example, with the choices $\kappa_t = t, \nu_t = 1 - t$, we get

$$u(z_t, t) = \sigma^{-1} \left[ -\sigma \sqrt{\frac{t}{1-t}} \epsilon + z_1 - z_0 - h_\theta(z_t, t) \right] \quad (16)$$

See section J for details. We write a generalized loss based on the ELBO as

$$\mathbb{E}_{p(t)p(x_1, z_0)p_\theta(z_1|x_1)p(z_t|z_1, z_0)} \left[ -\ln p_\theta(x_1|z_1) + \frac{\beta_t}{2} \left\| \sigma \sqrt{\frac{t}{1-t}} \epsilon + z_1 - z_0 - h_\theta(z_t, t) \right\|^2 \right] \quad (17)$$

Where $\beta_t$ (discussed further in section 4) is a relative weighting term, similar in spirit to $\beta$-VAE (Higgins et al., 2017; Alemi et al., 2018), allowing empirical re-balancing for metrics of interest, e.g. FID. Above loss is reminiscent of the SI training objective, but with an additional reconstruction term and the interpolants $z_t$ arising from the variational posterior. We use this training objective for all the experiments, and optimize it using stochastic gradient descent to jointly train all three components – encoder $p_\theta(z_1|x_1)$, decoder $p_\theta(x_1|z_1)$ and latent SI model $h_\theta(z_t, t)$. Note that we choose $p_\theta(x_1|z_1)$ to be a conditional gaussian in all experiments, resulting in a simple $L_2$ decoder loss.

**Observation-space stochastic interpolants:** To elucidate the connection with observation-space SI (Albergo et al., 2023) we derive the corresponding training objective in our framework, yielding:

$$\mathbb{E}_{p(t)p(x_1, x_0)p(x_t|x_1, x_0)} \left[ \frac{\beta_t}{2} \left\| \sigma \sqrt{\frac{t}{1-t}} \epsilon + x_1 - x_0 - h_\theta(x_t, t) \right\|^2 \right] \quad (18)$$

where $\beta_t$ has the same interpretation as in eq. (17), with $\beta_t = \sigma^{-2}$ corresponding to exact ELBO. See Section B for detailed proof. Comparing with the LSI loss (eq. (17)), the observation-space ELBO is precisely the LSI objective with the reconstruction term $-\ln p_\theta(x_1|z_1)$ removed and $z$ replaced by $x$. LSI recovers observation-space stochastic interpolants when the encoder and decoder are identity functions. All parameterizations (Section 4) and sampling procedures (Section 5) apply directly with $z$ replaced by $x$. Lastly, the likelihood control property of the above objective is trivially established – the objective corresponds to $\mathrm{KL}(\mathbb{Q}\|\mathbb{P}_\theta)$ for $\beta_t = \sigma^{-2}$ and $\mathrm{KL}(p_1\|p_\theta) \leq \mathrm{KL}(\mathbb{Q}\|\mathbb{P}_\theta)$ (eq. (41)), where $p_1$ is the true data distribution and $p_\theta$ is the data likelihood under the model.

**Learnable priors:** When the prior $p_0$ is parameterized (e.g., $p_\theta(z_0) = \mathcal{N}(\mu_\theta, \Sigma_\theta)$), the default construction above uses the same learnable prior for both processes ($q_0 = p_\theta$), so $\mathrm{KL}(q_0\|p_0) = 0$ and the ELBO retains the same form. The prior parameters are still learned: they affect the distribution of $z_0$ in the path integral $\mathbb{E}_\mathbb{Q}[\int \|u\|^2 dt]$, and gradients flow through $z_0 \sim p_\theta(z_0)$ via the reparameterization trick. Alternatively, if the variational process uses a fixed reference $q_0 \neq p_\theta$, the $\mathrm{KL}(q_0\|p_\theta)$ term appears as an additional regularizer penalizing deviation from the reference. Same carries over to the observation-space stochastic interpolants as well.

## 4 PARAMETERIZATION

Directly using the loss in eq. (17) leads to high variance in gradients and unreliable training due to the $\sqrt{1-t}$ in the denominator of the second term. Consequently, we consider several alternative parameterizations for the second term, including denoising and noise prediction (see section C for details). Among the alternatives considered, we found the following parameterization, referred to as InterpFlow, to reliably lead to better results and we use it in all our experiments.

$$\frac{\beta_t}{2} \left\| -\sigma \sqrt{t} \epsilon + \sqrt{1-t}(z_1 - z_0) + \sqrt{t} z_t - \hat{h}_\theta(z_t, t) \right\|^2 \quad (19)$$

Where $\hat{h}_\theta(z_t, t) \equiv \sqrt{t}z_t + \sqrt{1-t}\,h_\theta(z_t, t)$ and $\beta_t \equiv \beta/(1-t)$ is a time $t$ dependent weighting term, with $\beta$ a constant. Instead of explicitly using the weights $\beta_t$, due to $1-t$ in the denominator, we consider a change of variable for $t$ with the parametric family $t(s) = 1 - (1-s)^c$ with $s \sim \mathcal{U}[0,1]$ uniformly sampled. It can be shown that $p(t) \propto (1-t)^{\frac{1}{c}-1}$, therefore the change of variable provides the reweighting and we simply set $\beta_t = \beta$, a constant. Empirically, we found that a value of $c = 1$ (i.e. a uniform schedule) works the best for all parameterizations during training and sampling, except for NoisePred and Denoising, which preferred $c \approx 2$ during sampling. $c < 1$ led to degradation in FID. Figure 4 in appendix visualizes $t(s)$ for various values of $c$. While the ELBO suggests using $\beta = 1/\sigma^2$, we compute the two terms in eq. (17) as averages and experiment with different weightings. When used with optimizers like Adam or AdamW, $\beta$ can be interpreted as the relative weighting of the gradients from the two terms for the encoder $p_\theta(z_1|x_1)$. A lower value of $\beta$ leads the encoder to focus purely on the reconstruction and is akin to using a pre-trained encoder-decoder pair as $\beta \to 0$. A higher value of $\beta$ forces the encoder to adapt its representation for the second term as well. We empirically study the effect of $\beta$ in the experiments.

## 5 SAMPLING

For the InterpFlow parameterization, the learned drift $\hat{h}_\theta(z_t, t)$ is related to the original drift $h_\theta(z_t, t)$ as $h_\theta(z_t, t) = (\hat{h}(z_t, t) - \sqrt{t}z_t)/\sqrt{1-t}$ (see section F.2). We can sample from the model by discretizing the SDE in eq. (1), where $\sigma_t = \sigma$ for the choices of $\kappa_t = t, \nu_t = 1 - t$. However, to derive a flexible family of samplers where we can independently tune the dispersion $\sigma$ without retraining, we exploit Corollary 1 from Singh & Fischer (2024) to introduce a family of SDEs with the same marginal distributions as that for eq. (1)

$$dz_t = \left[ h_\theta(z_t, t) - \frac{(1-\gamma_t^2)\sigma^2}{2}\nabla_{z_t} \ln p_t(z_t) \right] dt + \gamma_t \sigma dw_t \tag{20}$$

Where $\gamma_t \geq 0$ can be chosen to control the amount of stochasticity introduced into sampling. For example, setting $\gamma_t = 0$ yields the probability flow ODE for deterministic sampling. In general, to use eq. (20) for $\gamma_t \neq 1$, the score function $\nabla_{z_t} \ln p_t(z_t)$ is needed as well. For the interpolant $z_t = \sigma\sqrt{t(1-t)}\epsilon + tz_1 + (1-t)z_0$, the score can be estimated using

$$\nabla_{z_t} \ln p_t(z_t) = -\frac{\mathbb{E}[\epsilon|z_t]}{\sigma\sqrt{t(1-t)}} \tag{21}$$

See section E for the proof. However, for Gaussian $z_0$, score can be computed from the drift $h_\theta(z_t, t)$ (Singh & Fischer, 2024) as following (see section D for details)

$$\nabla_x \ln p_t(z_t) = -z_t + th_\theta(z_t, t) \tag{22}$$

Section F provides detailed derivation of samplers for various parameterizations. For classifier free guided sampling (Ho & Salimans, 2022; Xie et al., 2024; Dao et al., 2023; Zheng et al., 2023; Singh & Fischer, 2024), we define the guided drift as a linear combination of the conditional drift $h_\theta(z_t, t, c)$ and the unconditional drift $h_\theta(z_t, t, c = \varnothing)$ as

$$h^{\text{cfg}}(z_t, t, c) \equiv (1 + \lambda)h_\theta(z_t, t, c) - \lambda h_\theta(z_t, t, c = \varnothing) \tag{23}$$

where $\lambda$ is the relative weight of the guidance, $c$ is the conditioning information and $c = \varnothing$ denotes no conditioning. Note that $\lambda = -1$ corresponds to unconditional sampling, $\lambda = 0$ corresponds to conditional sampling and $\lambda > 0$ further biases towards the modes of the conditional distribution.

## 6 EXPERIMENTS

We evaluate LSI on the standard ImageNet (2012) dataset (Deng et al., 2009; Russakovsky et al., 2015). We train models at various image resolutions and compare their sample quality using the Frechet Inception Distance (FID) metric (Heusel et al., 2017) for class conditional samples. All models were trained for 1000 epochs, except for the comparison in table 1 which reports FID at 2000 epochs. All results use deterministic sampler, using $\gamma_t = 0$, unless otherwise specified. A key implementation detail to note is that the encoder uses normalization and `tanh` to bound the scale of the latents. See sections O and P for additional details.

Table 1: **LSI enables joint learning for SI and cheaper sampling:** The latent space models achieve FID similar to observation space models of comparable size. However, the latent space model L has fewer parameters (reported in millions (M)) and FLOPs (reported in Giga (G)), as part of the parameters live in the encoder E and the decoder D. During sampling, encoder is not used, decoder is used only once, while the latent model L is run repeatedly, once for each sampling step. Therefore, FLOP savings from a computationally cheaper latent model accumulate with sampling steps.

| Resolution | FID @ 2K epochs | | # Params (M) | | Flops (G) | |
| | Latent | Observ. | Latent (E/D/L) | Observ. | Latent (E/D/L) | Observ. |
| --- | --- | --- | --- | --- | --- | --- |
| $64 \times 64$ | 2.62 | 2.57 | 392 (5/5/382) | 398 | 15/15/161 | 201 |
| $128 \times 128$ | 3.12 | 3.46 | 392 (5/5/382) | 400 | 59/59/327 | 466 |
| $256 \times 256$ | 3.91 | 3.87 | 393 (5/5/383) | 405 | 240/240/450 | 1288 |

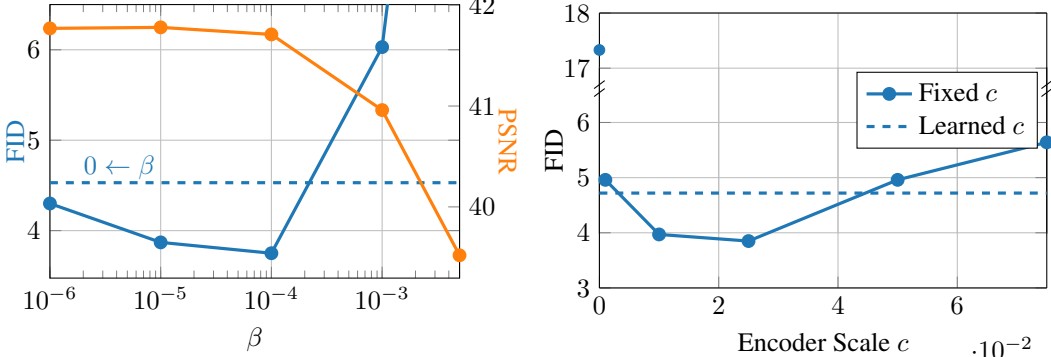

Figure 1: **Effect of loss trade-off $\beta$ and encoder noise scale $c$:** In the left panel, we evaluate the effect of loss trade-off weight $\beta$ for $128 \times 128$ models and observe that FID improves with $\beta$, until the degradation in reconstruction quality (PSNR) starts degrading FID. In the right panel, we evaluate the effect of encoder noise scale on FID. We also plot the FID for a model with learned scale as dashed line. A deterministic encoder performs the worst ($c = 0$), with FID improving with $c$ until it degrades again. Encoder with learned $c$ (dashed line) is outperformed by fixed $c$ in our experiments.

**LSI enables joint learning for SI :**   While SI doesn't allow latent variables, LSI enables joint learning of Encoder (E), Decoder (D), and Latent SI models (L). In table 1 we compare FID across various resolutions for LSI models against SI models trained directly in observation (pixel) space. LSI models achieve FIDs similar to the observation space models indicating on par performance in terms of the final FID. Models for both were chosen with similar architecture and number of parameters and trained for 2000 epochs. Reference comparison with other methods is provided in section R.

**LSI enables computationally cheaper sampling:**   In table 1 we also report the parameter counts (in millions) as well as FLOPs (in Giga) for the observation space SI model as well as E, D and L models for the LSI. For the latent L model, FLOPs are reported for a single forward pass. First note that the parameters in LSI are partitioned across the encoder E, the decoder D and the latent L models. At sampling time, encoder is not used, decoder is used only once, while the latent model is run multiple times, once for each step of sampling. Therefore, while the overall FLOP count for LSI and Observation space SI models is similar for a single forward pass, sampling with multiple steps becomes significantly cheaper. For example, sampling with 100 steps leads to 73.6% reduction in FLOPs for sampling $128 \times 128$ images and 48.6% for $256 \times 256$ images.

**Joint learning is beneficial:**   In fig. 1(left panel) we plot the FID as the weighting term $\beta$ is varied (eq. (19)). A higher $\beta$ forces the encoder to adapt the latents more for the second term of the loss. We observe that FID improves as $\beta$ increases, going from 4.53 (for $\beta \to 0$) to 3.75 ($\approx 17\%$ improvement) for $\beta = 0.0001$, indicating that this adaptation is beneficial for the overall performance. Eventually, FID worsens as $\beta$ is increased further. We also plot the reconstruction PSNR for each of these models

Table 2: **Joint training helps mitigate capacity shift:** We evaluate the effect of moving first $k$ and last $k$ convolutional blocks from the latent model L to encoder and decoder respectively, for $128 \times 128$ resolution models. This results in the overall parameter count staying roughly the same, but the number of FLOPs required for sampling changing significantly. We observe that the model trained with $\beta > 0$ perform better and maintains FID well, in comparison to the independently trained model ($\beta \to 0$), even when capacity is shifted away from the latent model L, resulting in $8.5\%$ reduction in FLOPs for sampling from $k = 0$ to $k = 6$.

| $k$ | FID ($\beta > 0$) | FID ($\beta \to 0$) | #Params. (E/D/L) | FLOPs (E/D/L) |
|---|---|---|---|---|
| 0 | 3.76 | 4.31 | 392 (5/5/382) | 59/59/327 |
| 3 | 3.91 | 4.55 | 389 (9/8/372) | 68/66/313 |
| 6 | 3.96 | 4.87 | 387 (13/12/362) | 75/73/299 |
| 9 | 4.61 | 4.98 | 383 (16/16/351) | 82/80/284 |

in orange and observe that increasing $\beta$ essentially trades-off reconstruction quality with generative performance. For too large a $\beta$, poor reconstruction quality leads to worsening FID. The dashed line indicates the performance when the encoder-decoder are trained independently of the latent model, limit of $\beta \to 0$. We implement it as a stop gradient operation in implementation, where the gradients from the second term of the loss are not backpropagated into $z_1$. To further assess the benefits of joint training, in table 2 we compare the FIDs between jointly trained model ($\beta > 0$) and independently trained model ($\beta \to 0$) as parameters are shifted from the latent model L to the encoder E and decoder D models, by moving first $k$ and last $k$ convolutional blocks from the latent model to the encoder and the decoder respectively. While this keeps the total parameter count roughly the same, the number of FLOPs required for sampling changes significantly. The jointly trained model performs better and maintains FID well even when capacity shifts away from the latent model, resulting in $8.5\%$ reduction in FLOPs required for sampling from $k = 0$ to $k = 6$.

**Encoder noise scale affects performance:** The stochasticity of the encoder $p_\theta(z_1|x)$ has a significant impact on the performance. We parameterize the encoder as a conditional Gaussian $N(z_1; \mu_\theta(x), \Sigma_\theta(x))$ where $\Sigma(x)$ is assumed to be diagonal. We experimented with a purely deterministic encoder ($\Sigma_\theta(x) = 0$), learned $\Sigma_\theta(x)$ and constant noise $\Sigma_\theta(x) = cI$. In fig. 1(right panel) we plot FID as the encoder output stochasticity $c$ is varied. Dashed line indicates performance with learned $\Sigma_\theta(x)$. A deterministic encoder ($c = 0$) performs poorly. FID improves as the noise scale $c$ is increased, until eventually it degrades again. While learned $\Sigma_\theta(x)$ (dashed line) performs well, fixed $c$ models achieved higher FID.

**InterpFlow parameterization performs better than alternatives:** In table 3 we compare different parameterizations discussed in section 4 and section C. The InterpFlow parameterization consistently led to better FID. Both OrigFlow and NoisePred parameterizations exhibited higher variance gradients and noisy optimization. While Denoising parameterization resulted in less noisy training, InterpFlow parameterization led to fastest improvement in FID.

**LSI supports diverse $p_0$:** In table 4 we report FID achieved by LSI using different prior $p_0(z_0)$ distributions. While Gaussian $p_0$ performs the best, other choices for $p_0$ yield competitive results indicating that LSI retains one of the key strengths of SI – support for diverse $p_0$ distributions. See section N for additional details. To allow flexible sampling using eq. (20), we modified latent SI model to output extra output channels and augmented the loss with another term to estimate $\mathbb{E}[\epsilon|z_t]$. Equation (21) was used to compute the score and sample with the deterministic sampler using $\gamma_t = 0$.

**LSI supports flexible sampling:** In fig. 2 and fig. 3 we qualitatively demonstrate flexible sampling with LSI model for popular use cases. Figure 2 demonstrates compatibility of classifier free guidance (CFG) with LSI, using eq. (22). Increasing guidance weight $\lambda$ results in more typical samples. First $z_0$ is sampled from $p_0(z_0)$, Gaussian in this example, following which eq. (20) is simulated forward in time, using class conditional drift with different guidance weights $\lambda$. In fig. 3 a given 'Original' image (shown leftmost) is first encoded to yield it's representation $z_1$, which is then inverted by simulating probability flow ODE (setting $\gamma_t = 0$ in eq. (20)) backward in time from $t = 1$ to $t = 0$, yielding $z_0$ (similar to DDIM inversion (Song et al., 2020a)). Using this $z_0$ as starting point, eq. (20)

Table 3: **Effect of parameterization:** We compare various parameterization schemes at $128 \times 128$ resolution. $\mathrm{InterpFlow}$ parameterization performs better against the alternatives.

| Parameterization | FID @1K epochs |
|---|---|
| OrigFlow | 4.56 |
| NoisePred | 4.73 |
| Denoising | 4.28 |
| InterpFlow | 3.76 |

Table 4: **LSI supports diverse $p_0$:** LSI retains one of the key strengths of SI – support for arbitrary $p_0$ distribution. Different $p_0$ achieve competetive FID for $128 \times 128$ resolution model.

| $p_0$ | FID @1K epochs |
|---|---|
| Uniform | 4.81 |
| Laplacian | 4.45 |
| Gaussian | 3.76 |
| Gaussian Mixture | 4.26 |

| $\lambda = 0.$ | $\lambda = 1.$ | $\lambda = 3.$ | $\lambda = 5.$ | $\lambda = 0.$ | $\lambda = 1.$ | $\lambda = 3.$ | $\lambda = 5.$ |
|---|---|---|---|---|---|---|---|

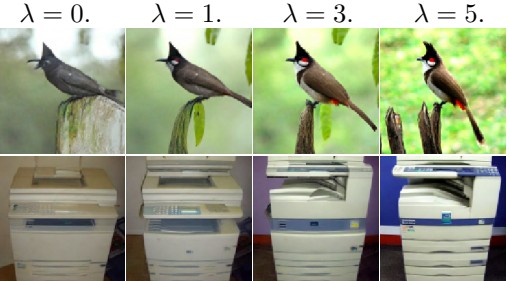 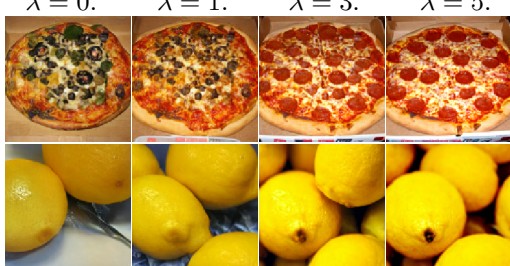

Figure 2: **LSI supports CFG sampling.** Class conditional samples are visualized with increasing guidance weight $\lambda$ leading to more typical samples for the class. See text for details.

is simulated forward is time using $\gamma_t \equiv \gamma(1 - t)$ for different values of $\gamma$. We show three samples for each value of $\gamma$ and observe increasing diversity with increasing $\gamma$. See section Q for additional details and results.

# 7 RELATED WORK

Latent Stochastic Interpolants (LSI) draw from insights in diffusion models, latent variable models, and continuous-time generative processes. We discuss key works from these areas in the following.

**Diffusion Models:** Diffusion models, originating from foundational work on score matching (Vincent, 2011; Song & Ermon, 2019) and early variational formulation (Sohl-Dickstein et al., 2015), gained prominence with Denoising Diffusion Probabilistic Models (DDPMs) (Ho et al., 2020). Subsequent improvements focused on architectural choices and learned variances (Nichol & Dhariwal, 2021), faster sampling via Denoising Diffusion Implicit Models (DDIMs) (Song et al., 2020a), progressive distillation (Salimans & Ho, 2022), and powerful conditional generation through techniques like classifier-free guidance (Ho & Salimans, 2022). Further exploration of the design space (Karras et al., 2022; 2024) has lead to highly performant models. More recently, diffusion inspired consistency models (Song et al., 2023) have emerged, offering efficient generation. LSI complements these with a flexible method for jointly learning in a latent space using richer prior distributions.

**Latent Variable Models and Expressive Priors:** Variational Autoencoders (VAEs) (Kingma et al., 2013; Rezende et al., 2014) learn a compressed representation $z$ of data $x$, but are limited by the expressiveness of the prior $p(z)$ (NVAE (Vahdat & Kautz, 2020), LSGM (Vahdat et al., 2021)), as they typically use simple priors (e.g., isotropic Gaussian). LSI addresses this by jointly learning a flexible generative process in the latent space, enabling powerful transformations of the simple prior. Early work (Sohl-Dickstein et al., 2015) derived ELBO for discrete time diffusion models, while Variational Diffusion Models (VDM) (Kingma et al., 2021) interpret diffusion models as a specific type of VAE with Gaussian noising process. In contrast, while LSI also optimizes an ELBO, it allows for a broader choice of the prior $p(z_0)$ and the transforms mapping the prior to the learned aggregated posterior. Our work is similar in spirit to models like NVAE, which employed deep hierarchical latent representations, and LSGM, which proposed training score-based models in the latent space of a VAE, but offers a flexible framework similar to SI allowing a rich family of priors and latent space

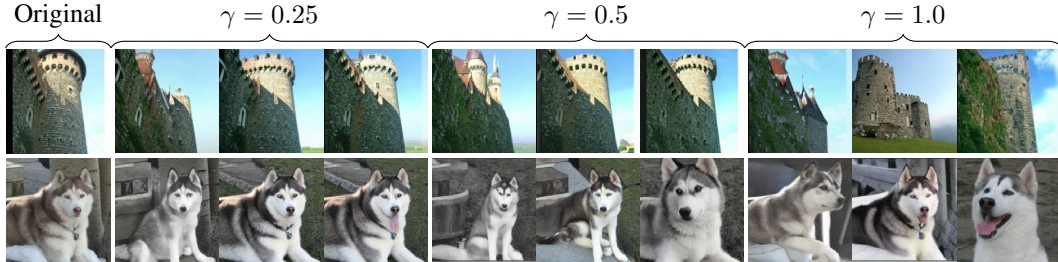

Figure 3: **LSI supports flexible sampling.** We demonstrate inversion of an 'Original' image, using reverse probability flow ODE (similar to DDIM inversion), followed by forward stochastic sampling to yield samples similar to it, with diversity increasing with $\gamma$ (eq. (20)). See text for details.

dynamics. Note that LDM (Rombach et al., 2022) train a diffusion generative model in the latent space of a *fixed* encoder-decoder pair – making their latents actually *observed* from the point of view of generative modeling.

**Continuous-Time Generative Processes:** While diffusion models have been formulated and studied using continuous time dynamics (Song et al., 2020b;a; Kingma et al., 2021; Vahdat et al., 2021), their relation to Continuous Normalizing Flows (CNFs) (Chen et al., 2018; Grathwohl et al., 2019) offers another perspective on continuous-time transformations. Early training challenges with the CNFs have been addressed by newer methods like Flow Matching (FM) (Lipman et al., 2022; Xu et al., 2022), Conditional Flow Matching (CFM) (Neklyudov et al., 2023; Tong et al., 2023), and Rectified Flow (Liu et al., 2022). These approaches propose simulation-free training by regressing vector fields of fixed conditional probability paths. However, likelihood control is typically not possible (Albergo et al., 2023), consequently extension to jointly learning in latent space is ill-specified. In contrast, LSI optimizes an ELBO, offering likelihood control along with joint learning in a latent space. Stochastic Interpolants (SI) (Albergo et al., 2023) provides a unifying perspective on generative modeling, capable of bridging *any* two probability distributions via a continuous-time stochastic process, encompassing aspects of both flow-based and diffusion-based methods. While SI formulates learning the velocity field and score function directly in the observation space using pre-specified stochastic interpolants, LSI arrives at a similar objective in the latent space, as part of the ELBO, from the specific choices of the approximate variational posterior. LSI reduces to SI when encoder and decoder are chosen to be Identity functions. SI is related to the Optimal Transport and the Schrödinger Bridge problem (SBP) which have been explored as a basis for generative modeling (De Bortoli et al., 2021; Wang et al., 2021; Shi et al., 2023). While LSI learns a transport, its primary objective is data log-likelihood maximization via the ELBO, rather than solving a specific OT or SBP.

## 8 CONCLUSION

In this paper, we introduced Latent Stochastic Interpolants (LSI), generalizing Stochastic Interpolants to enable joint end-to-end training of an encoder, a decoder, and a generative model operating entirely within the learned latent space. LSI overcomes the limitation of simple priors of the normal diffusion models and mitigates the computational demands of applying SI directly in high-dimensional observation spaces, while preserving the generative flexibility of the SI framework. LSI leverage SDE-based Evidence Lower Bound to offer a principled approach for optimizing the entire model. We validate the proposed approach with comprehensive experimental studies on standard ImageNet benchmark. Our method offers scalability along with a unifying perspective on continuous-time generative models with dynamic latent variables. However, to achieve scalable training, our approach makes simplifying assumptions for the variational posterior approximation. While restrictive, and common with other methods, these assumptions do not seem to limit the empirical performance.

## ACKNOWLEDGMENTS

We would like to thank Kevin J. Shih and Ian Fischer for proofreading early drafts of this manuscript and providing valuable feedback.

## REPRODUCIBILITY STATEMENT

We have included detailed proofs of all the key theoretical results in the appendix. Sections 6 and O provide key training and evaluation setup details. Section P provides the necessary architecture details to reproduce the models used in the experiments. Section Q provides additional sampling setup details.

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

APPENDIX

## A  PROOF OF ELBO FOR DYNAMIC LATENT VARIABLES (EQ. (3))

We provide a self-contained proof of the variational lower bound stated in eq. (3). The proof is based on the approach of Li et al. (2020), but in a more general form. We first establish the path-space KL divergence between two diffusion processes via Girsanov's theorem (Theorem 1), then use it to derive the ELBO (Theorem 2). Let $\mathbb{P}_\theta$ be the path measure under the model as in eq. (1) and let $\mathbb{Q}$ be the path measure under the variational posterior process approximation as in eq. (2).

**Theorem 1 (Path-space KL via Girsanov's theorem):** Consider two SDEs, starting at the same starting point $\tilde{z}_0 = z_0$, sharing the same dispersion coefficient $\sigma(z_t, t)$ but potentially different initial distributions — $z_0 \sim q_0(z_0)$ for $\mathbb{Q}$ and $z_0 \sim p_0(z_0)$ for $\mathbb{P}_\theta$:

$$d\tilde{z}_t = h_\theta(\tilde{z}_t, t)\, dt + \sigma(\tilde{z}_t, t)\, dw_t, \qquad \text{(model, path measure } \mathbb{P}_\theta) \qquad \text{(eq. 1)}$$

$$dz_t = h_\phi(z_t, t)\, dt + \sigma(z_t, t)\, dw_t, \qquad \text{(variational, path measure } \mathbb{Q}) \qquad \text{(eq. 2)}$$

Define $u(z_t, t)$ by $\sigma(z_t, t)\, u(z_t, t) = h_\phi(z_t, t) - h_\theta(z_t, t)$ (eq. (5)). Then:

$$\mathrm{KL}(\mathbb{Q}\|\mathbb{P}_\theta) = \mathrm{KL}(q_0\|p_0) + \frac{1}{2}\, \mathbb{E}_\mathbb{Q}\left[\int_0^1 \|u(z_t, t)\|^2\, dt\right] \qquad (24)$$

When $q_0 = p_0$ (both processes share the same initial distribution, potentially learnable), the initial KL vanishes and the path-space KL reduces to the dynamics mismatch alone.

*Proof.* The full path measure factorizes as the initial distribution times the conditional path measure given the initial state. The Radon-Nikodym derivative therefore decomposes as

$$\frac{d\mathbb{Q}}{d\mathbb{P}_\theta}(\mathcal{Z}) = \frac{q_0(z_0)}{p_0(z_0)} \cdot \frac{d\mathbb{Q}^{z_0}}{d\mathbb{P}_\theta^{z_0}}(\mathcal{Z}) \qquad (25)$$

where $\mathbb{Q}^{z_0}$ and $\mathbb{P}_\theta^{z_0}$ denote the conditional path measures given the initial state $z_0$. Next we use Girsanov's theorem to evaluate the second factor. Under $\mathbb{Q}$, the process satisfies $dz_t = h_\phi\, dt + \sigma\, dw_t^\mathbb{Q}$ where $w_t^\mathbb{Q}$ is a standard Brownian motion. Define the $\mathbb{P}_\theta$-Brownian motion via $dw_t^{\mathbb{P}_\theta} = \sigma^{-1}(dz_t - h_\theta\, dt)$. Substituting the $\mathbb{Q}$-dynamics for $dz_t$:

$$dw_t^{\mathbb{P}_\theta} = dw_t^\mathbb{Q} + u(z_t, t)\, dt \qquad (26)$$

That is, under $\mathbb{Q}$, the process $w_t^{\mathbb{P}_\theta}$ acquires a drift $u_t$. By Girsanov's theorem:

$$\frac{d\mathbb{Q}^{z_0}}{d\mathbb{P}_\theta^{z_0}} = \exp\left(\int_0^1 u_t^T\, dw_t^{\mathbb{P}_\theta} - \frac{1}{2}\int_0^1 \|u_t\|^2\, dt\right) \qquad (27)$$

Substituting $dw_t^{\mathbb{P}_\theta} = dw_t^\mathbb{Q} + u_t\, dt$ and combining with the initial density ratio:

$$\ln \frac{d\mathbb{Q}}{d\mathbb{P}_\theta} = \ln \frac{q_0(z_0)}{p_0(z_0)} + \int_0^1 u_t^T\, dw_t^\mathbb{Q} + \frac{1}{2}\int_0^1 \|u_t\|^2\, dt \qquad (28)$$

Taking the expectation under $\mathbb{Q}$:

$$\mathrm{KL}(\mathbb{Q}\|\mathbb{P}_\theta) = \underbrace{\mathbb{E}_{q_0}\left[\ln \frac{q_0(z_0)}{p_0(z_0)}\right]}_{\mathrm{KL}(q_0\|p_0)} + \underbrace{\mathbb{E}_\mathbb{Q}\left[\int_0^1 u_t^T\, dw_t^\mathbb{Q}\right]}_{=\,0} + \frac{1}{2}\, \mathbb{E}_\mathbb{Q}\left[\int_0^1 \|u_t\|^2\, dt\right] \qquad (29)$$

The Itô integral $\int_0^1 u_t^T\, dw_t^\mathbb{Q}$ is a martingale under $\mathbb{Q}$ (under the standard integrability condition $\mathbb{E}_\mathbb{Q}[\int_0^1 \|u_t\|^2\, dt] < \infty$), and the expectation of a martingale starting at zero is zero. $\square$

**Remark 1:** Theorem 1 applies to any two diffusion processes sharing the same dispersion, regardless of whether the state space represents latent variables or observations. In particular, it applies directly in observation space (with $z$ replaced by $x$).

**Remark 2 (Learnable prior):** When the prior $p_0$ is parameterized (e.g., $p_\theta(z_0) = \mathcal{N}(\mu_\theta, \Sigma_\theta)$), the natural construction uses the same learnable prior for both processes ($q_0 = p_\theta$), so $\text{KL}(q_0 \| p_0) = 0$ and the ELBO retains the same form. The prior parameters are still learned: they affect the distribution of $z_0$ in the path integral $\mathbb{E}_\mathbb{Q}[\int \|u\|^2 \, dt]$, and gradients flow through $z_0 \sim p_\theta(z_0)$ via the reparameterization trick. Alternatively, if the variational process uses a fixed reference $q_0 \neq p_\theta$, the $\text{KL}(q_0 \| p_\theta)$ term appears as an additional regularizer penalizing deviation from the reference.

**Theorem 2 (ELBO for dynamic latent variables, eqs. (3) and (5)):** Under the setup of Theorem 1, with potentially learnable initial distributions $p_0$ and $q_0$, let $x_{t_i}$ be observations at times $t_i \in [0, 1]$, $i = 1, \ldots, n$, assumed to depend on the latent state only through $z_{t_i}$, i.e., $p_\theta(x_{t_i}|\mathcal{Z}) = p_\theta(x_{t_i}|z_{t_i})$. Then:

$$\ln p_\theta(x_{t_1}, \ldots, x_{t_n}) \geq \mathbb{E}_\mathbb{Q}\left[\sum_{i=1}^n \ln p_\theta(x_{t_i}|z_{t_i}) - \ln \frac{q_0(z_0)}{p_0(z_0)} - \frac{1}{2}\int_0^1 \|u(z_t, t)\|^2 \, dt\right] \tag{30}$$

$$= \mathbb{E}_\mathbb{Q}\left[\sum_{i=1}^n \ln p_\theta(x_{t_i}|z_{t_i})\right] - \text{KL}(\mathbb{Q}\|\mathbb{P}_\theta) \tag{31}$$

When $q_0 = p_0$, the second term on the right vanishes and we recover the special case of above as stated in Li et al. (2020).

*Proof.* Under the model, the latent path evolves according to $\mathbb{P}_\theta$ (eq. (1)) and observations are generated conditionally at each time $t_i$. The marginal likelihood is obtained by integrating over all latent paths:

$$p_\theta(x_{t_1}, \ldots, x_{t_n}) = \int \prod_{i=1}^n p_\theta(x_{t_i}|z_{t_i}) \, d\mathbb{P}_\theta(\mathcal{Z}) \tag{32}$$

Since both $\mathbb{P}_\theta$ and $\mathbb{Q}$ share the same dispersion and initial distribution, they are mutually absolutely continuous (by Girsanov's theorem, under standard regularity). We can therefore re-express the integral using the variational path measure $\mathbb{Q}$ (eq. (2)) as a proposal:

$$p_\theta(x_{t_1}, \ldots, x_{t_n}) = \mathbb{E}_\mathbb{Q}\left[\prod_{i=1}^n p_\theta(x_{t_i}|z_{t_i}) \cdot \frac{d\mathbb{P}_\theta}{d\mathbb{Q}}(\mathcal{Z})\right] \tag{33}$$

Taking the logarithm of both sides and using Jensen's inequality:

$$\ln p_\theta(x_{t_1}, \ldots, x_{t_n}) \geq \mathbb{E}_\mathbb{Q}\left[\sum_{i=1}^n \ln p_\theta(x_{t_i}|z_{t_i}) + \ln \frac{d\mathbb{P}_\theta}{d\mathbb{Q}}(\mathcal{Z})\right] \tag{34}$$

From the proof of Theorem 1 (with $q_0 = p_0$, so the initial density ratio cancels), the log Radon-Nikodym derivative expressed in terms of the $\mathbb{Q}$-Brownian motion is:

$$\ln \frac{d\mathbb{P}_\theta}{d\mathbb{Q}}(\mathcal{Z}) = -\ln \frac{q_0(z_0)}{p_0(z_0)} - \int_0^1 u_t^T \, dw_t^\mathbb{Q} - \frac{1}{2}\int_0^1 \|u_t\|^2 \, dt \tag{35}$$

Substituting the above we get:

$$\ln p_\theta(x_{t_1}, \ldots, x_{t_n}) \geq \mathbb{E}_\mathbb{Q}\left[\sum_{i=1}^n \ln p_\theta(x_{t_i}|z_{t_i}) - \ln \frac{q_0(z_0)}{p_0(z_0)} - \int_0^1 u_t^T \, dw_t^\mathbb{Q} - \frac{1}{2}\int_0^1 \|u_t\|^2 \, dt\right] \tag{36}$$

$$= \mathbb{E}_\mathbb{Q}\left[\sum_{i=1}^n \ln p_\theta(x_{t_i}|z_{t_i}) - \ln \frac{q_0(z_0)}{p_0(z_0)} - \frac{1}{2}\int_0^1 \|u_t\|^2 \, dt\right] \tag{37}$$

$$= \mathbb{E}_\mathbb{Q}\left[\sum_{i=1}^n \ln p_\theta(x_{t_i}|z_{t_i})\right] - \text{KL}(\mathbb{Q}\|\mathbb{P}_\theta) \tag{38}$$

Where the Itô integral $\int_0^1 u_t^T \, dw_t^\mathbb{Q}$ vanishes under $\mathbb{E}_\mathbb{Q}$ (as established in Theorem 1). $\square$

**Remark 3:** The bound has a natural interpretation: the first term is a reconstruction likelihood (how well the model explains observations given the latent path) and the second term penalizes the mismatch between the variational and model path distributions. $\frac{1}{2}\int_0^1 \|u_t\|^2$ can also be seen as the control cost required to steer the model process $\mathbb{P}_\theta$ to match the variational process $\mathbb{Q}$ (Theodorou, 2015). The bound is tight when $\mathbb{Q} = \mathbb{P}_\theta(\cdot \mid x_{t_1}, \ldots, x_{t_n})$, i.e., when the variational process equals the true posterior process.

## B  OBSERVATION-SPACE STOCHASTIC INTERPOLANTS

The LSI framework (Section 3) jointly trains an encoder, decoder, and latent generative model. Here we consider the special case where the generative process operates directly in observation space, without an encoder or decoder. This corresponds to setting $z_t \equiv x_t$ for all $t$, making the latent process identical to the observation process.

**Setup:**  The generative model is an SDE directly in observation space,

$$d\tilde{x}_t = h_\theta(\tilde{x}_t, t)\, dt + \sigma_t\, d\tilde{w}_t, \quad \tilde{x}_0 \sim p_0(x_0) \tag{39}$$

i.e., eq. (1) with $z \to x$. The prior $p_0(x_0)$ may be fixed or learnable (e.g., $p_\theta(x_0) = \mathcal{N}(\mu_\theta, \Sigma_\theta)$). The marginal at $t = 1$ defines the model distribution $p_\theta(x_1)$.

The variational process $\mathbb{Q}$ is constructed using the same diffusion bridge machinery as in Section 2.2, now bridging $p_0(x_0)$ and $p_1(x_1) = p_{\text{data}}(x_1)$ directly in observation space. As in the previous section, both $\mathbb{Q}$ and $\mathbb{P}_\theta$ can use potentially different initial distributions $q_0$ and $p_0$. Starting from the linear SDE $dx_t = h_t x_t\, dt + \sigma_t\, dw_t$ (cf. eq. (7)) with $z \to x$) and applying Doob's h-transform to condition on ending at $x_1 \sim p_1$, the drift is (cf. eq. (9)):

$$h_\phi(x_t, t) = h_t x_t + \sigma_t^2 \nabla_{x_t} \ln p(x_1 \mid x_t) \tag{40}$$

By construction, $\mathbb{Q}$ has marginal $p_{\text{data}} = p_1$ at $t = 1$, while $\mathbb{P}_\theta$ has marginal $p_\theta(x_1)$.

**ELBO:**  Since $p_{\text{data}} = p_1$ and $x_1$ is a deterministic function of the path $\mathcal{X} \sim \mathbb{Q}$, the data processing inequality gives

$$\mathrm{KL}(p_1 \| p_\theta) \leq \mathrm{KL}(\mathbb{Q} \| \mathbb{P}_\theta) \tag{41}$$

Expanding the left side and rearranging yields the evidence lower bound:

$$\mathbb{E}_{p_1}[\ln p_\theta(x_1)] \geq \mathbb{E}_{p_1}[\ln p_1(x_1)] - \mathrm{KL}(\mathbb{Q} \| \mathbb{P}_\theta) \tag{42}$$
$$= -\mathrm{H}[p_1] - \mathrm{KL}(\mathbb{Q} \| \mathbb{P}_\theta) \tag{43}$$

Where $\mathrm{H}[p_1]$ is the entropy of the data distribution $p_1$. Using Theorem 1, the path-space KL can be evaluated via Girsanov's theorem as (with $z \to x$):

$$\mathrm{KL}(\mathbb{Q} \| \mathbb{P}_\theta) = \mathrm{KL}(q_0 \| p_0) + \frac{1}{2} \mathbb{E}_{\mathbb{Q}}\left[ \int_0^1 \| u(x_t, t) \|^2\, dt \right] \tag{44}$$

where $\sigma_t\, u(x_t, t) = h_\phi(x_t, t) - h_\theta(x_t, t)$, as in eq. (5). If $q_0 = p_0$, the first term on the right vanishes, leaving only the dynamics cost. As in main text, we assume $q_0 = p_0$ in the following.

**Simulation-free training:**  All the simulation-free machinery from section 3 carries over with $z \to x$. Using the observation-space interpolant (cf. eq. (12)):

$$x_t = \eta_t \epsilon + \kappa_t x_1 + \nu_t x_0, \quad \epsilon \sim \mathcal{N}(0, I) \tag{45}$$

$u(x_t, t)$ takes the following form – similar to the result in section 3 (cf. eq. (15)):

$$u(x_t, t) = \sigma_t^{-1} \left[ \left( \frac{d\eta_t}{dt} - \frac{\sigma_t^2}{2\eta_t} \right) \epsilon + \frac{d\kappa_t}{dt} x_1 + \frac{d\nu_t}{dt} x_0 - h_\theta(x_t, t) \right] \tag{46}$$

**Final objective:**  Substituting (44) and (46) into (43), the observation-space ELBO is:

$$-\mathrm{H}[p_1] - \frac{1}{2} \mathbb{E}_{t \sim \mathcal{U}[0,1],\, x_1 \sim p_1,\, x_0 \sim p_0,\, \epsilon \sim \mathcal{N}(0,I)} \left[ \| u(x_t, t) \|^2 \right] \tag{47}$$

where the entropy $\mathrm{H}[p_1]$ of the data distribution is a constant and is independent of the model parameters. For the linear choice $\kappa_t = t$, $\nu_t = 1 - t$, this specializes to the following loss to be minimized (cf. eq. (17)):

$$\frac{\beta_t}{2} \mathbb{E}\left[ \left\| \sigma \sqrt{\frac{t}{1-t}} \epsilon + x_1 - x_0 - h_\theta(x_t, t) \right\|^2 \right] \tag{48}$$

where $\beta_t$ has the same interpretation as in eq. (17), of a generalized weighting term, and the constant term has been dropped.

**Remark 4:** Comparing with the LSI loss (eq. (17)), the observation-space ELBO is precisely the LSI objective with the reconstruction term $-\ln p_\theta(x_1 \mid z_1)$ removed and $z \to x$ throughout. This confirms the consistency of the framework: LSI reduces to observation-space stochastic interpolants when the encoder and decoder are identity functions. All parameterizations (Section 4) and sampling procedures (Section 5) apply directly with $z \to x$.

**Remark 5 (Learnable prior):** The ELBO in eq. (47) supports a learnable prior $p_\theta(x_0)$ without modification. If both $\mathbb{Q}$ and $\mathbb{P}_\theta$ start from the same $p_\theta(x_0)$, the initial KL vanishes regardless of the prior's form (Theorem 1). The prior parameters are still learned through the interpolant $x_t = \eta_t\epsilon + \kappa_t x_1 + \nu_t x_0$ and the target $\frac{d\nu_t}{dt}x_0$ in eq. (46), providing gradients via the reparameterization trick. Note that the interpolant coefficients $\eta_t, \kappa_t, \nu_t$ depend only on the base SDE parameters $h_t, \sigma_t$, not on $p_0$, so changing the prior affects only the sampling distribution of $x_0$ — not the interpolant structure.

## C  PARAMETERIZATIONS

For the linear choice of $\kappa_t = t, \nu_t = 1 - t$ (section J) used for experiments in this paper, the loss term with $u(z_t, t)$ is

$$\mathbb{E}_{t\sim\mathcal{U}[0,1]}\mathbb{E}_{p(x_1,z_0,z_1)}\mathbb{E}_{p(z_t|z_1,z_0)}\frac{1}{2\sigma^2}\left\|-\sigma\sqrt{\frac{t}{1-t}}\epsilon + z_1 - z_0 - h_\theta(z_t,t)\right\|^2 \tag{49}$$

Where $\epsilon \sim N(0, I)$. If $z_0$ is also Gaussian, $z_0 \sim N(0, I)$, we can combine $\epsilon, z_0$ to yield $z_t = tz_1 + \sqrt{(1-t)(\sigma^2 t + 1 - t)}z_0$ and rewrite the above as

$$\mathbb{E}_{t\sim\mathcal{U}[0,1]}\mathbb{E}_{p(x_1,z_0,z_1)}\mathbb{E}_{p(z_t|z_1,z_0)}\frac{1}{2}\left\|z_1 - \sqrt{\frac{\sigma^2 t + 1 - t}{1-t}}z_0 - h_\theta(z_t,t)\right\|^2 \tag{50}$$

Directly using above forms leads to high variance in gradients and unreliable training with frequent NaNs due to the $\sqrt{1-t}$ in the denominator. Consequently, we consider alternative parameterizations as discussed in the following. Two of the parameterizations $\mathrm{OrigFlow}$ and $\mathrm{InterpFlow}$ are applicable for arbitrary $p_0$, while the remaining two $\mathrm{Denoising}$ and $\mathrm{NoisePred}$ are applicable when $z_0$ is Gaussian. For each of these parameterizations, we also derive the corresponding sampler in section F

### C.1  OrigFlow

With straightforward manipulation of the term inside the expectation we arrive at

$$\frac{1}{2\sigma^2}\frac{1}{1-t}\left\|\sqrt{1-t}(z_1 - z_0) - \sigma\sqrt{t}\epsilon - \hat{h}_\theta(z_t,t)\right\|^2 \tag{51}$$

where $\hat{h}_\theta(z_t, t) \equiv \sqrt{1-t}h_\theta(z_t, t)$. We rewrite above in terms of a time dependent weighting $\beta_t \equiv \frac{1}{\sigma^2(1-t)}$ as following.

$$\frac{\beta_t}{2}\left\|\sqrt{1-t}(z_1 - z_0) - \sigma\sqrt{t}\epsilon - \hat{h}_\theta(z_t,t)\right\|^2 \tag{52}$$

When $z_0$ is Gaussian, we can rewrite as

$$\frac{\beta_t}{2}\left\|\sqrt{1-t}z_1 - \sqrt{\sigma^2 t + 1 - t}z_0 - \hat{h}_\theta(z_t,t)\right\|^2 \tag{53}$$

This objective can be viewed as estimating $\hat{h}_\theta(z_t, t) \equiv \mathbb{E}[\sqrt{1-t}z_1 - \sqrt{\sigma^2 t + 1 - t}z_0|z_t]$ with a time $t$ dependent weighting $\beta_t$.

## C.2 InterpFlow

Again, starting with the loss term with $u(z_t, t)$ and straightforward manipulations we arrive at the parameterization

$$\frac{1}{2\sigma^2} \left\| -\sigma\sqrt{\frac{t}{1-t}}\epsilon + z_1 - z_0 - h_\theta(z_t, t) \right\|^2 \tag{54}$$

$$= \frac{1}{2\sigma^2} \left\| -\sigma\sqrt{\frac{t}{1-t}}\epsilon + z_1 - z_0 + \sqrt{\frac{t}{1-t}}z_t - \sqrt{\frac{t}{1-t}}z_t - h_\theta(z_t, t) \right\|^2 \tag{55}$$

$$= \frac{\beta_t}{2} \left\| -\sigma\sqrt{t}\epsilon + \sqrt{1-t}(z_1 - z_0) + \sqrt{t}z_t - \hat{h}_\theta(z_t, t) \right\|^2 \tag{56}$$

Where $\hat{h}_\theta(z_t, t) \equiv \sqrt{t}z_t + \sqrt{1-t}h_\theta(z_t, t)$ and $\beta_t \equiv \frac{1}{\sigma^2(1-t)}$. To gain insights into this parameterization, let's consider the term inside the norm and substitute $z_t$

$$- \sigma\sqrt{t}\epsilon + \sqrt{1-t}(z_1 - z_0) + \sqrt{t}z_t \tag{57}$$

$$= -\sigma\sqrt{t}\epsilon + \sqrt{1-t}(z_1 - z_0) + \sqrt{t}(tz_1 + (1-t)z_0 + \sigma\sqrt{t(1-t)}\epsilon) \tag{58}$$

$$= (\sqrt{1-t} + t\sqrt{t})z_1 + (\sqrt{t}(1-t) - \sqrt{1-t})z_0 + \sigma(t\sqrt{1-t} - \sqrt{t})\epsilon \tag{59}$$

Leading to

$$\frac{\beta_t}{2} \left\| (\sqrt{1-t} + t\sqrt{t})z_1 + (\sqrt{t}(1-t) - \sqrt{1-t})z_0 + \sigma(t\sqrt{1-t} - \sqrt{t})\epsilon - \hat{h}_\theta(z_t, t) \right\|^2 \tag{60}$$

The term $(\sqrt{1-t} + t\sqrt{t})z_1 + (\sqrt{t}(1-t) - \sqrt{1-t})z_0 + \sigma(t\sqrt{1-t} - \sqrt{t})\epsilon$ reduces to $z_1 - z_0$ at $t = 0$ and $z_1 - \sigma\epsilon$ at $t = 1$. Since this term appears to interpolate between the two, we refer to this parameterization as InterpFlow. When $z_0$ is also Gaussian, we can combine $\epsilon, z_0$ and rewrite as

$$\frac{\beta_t}{2} \left\| (\sqrt{1-t} + t\sqrt{t})z_1 + (\sqrt{t(1-t)} - 1)\sqrt{\sigma^2 t + 1 - t}z_0 - \hat{h}_\theta(z_t, t) \right\|^2 \tag{61}$$

Observe that, with $\sigma = 1$, the term $(\sqrt{1-t} + t\sqrt{t})z_1 + (\sqrt{t(1-t)} - 1)z_0$ reduces to $z_1 - z_0$ both at $t = 0$ and $t = 1$.

## C.3 Denoising

This parameterization is applicable only when $z_0$ is Gaussian. Starting with the loss term with $u(z_t, t)$ and using the fact that $z_t = tz_1 + \sqrt{(1-t)(\sigma^2 t + 1 - t)}z_0$, we can manipulate the objective as following

$$\frac{1}{2} \left\| z_1 - \sqrt{\frac{\sigma^2 t + 1 - t}{1-t}}z_0 - h_\theta(z_t, t) \right\|^2 \tag{62}$$

$$= \frac{1}{2} \left\| z_1 - \sqrt{\frac{\sigma^2 t + 1 - t}{1-t}}\frac{z_t - tz_1}{\sqrt{(1-t)(\sigma^2 t + 1 - t)}} - h_\theta(z_t, t) \right\|^2 \tag{63}$$

$$= \frac{1}{2} \left\| z_1 - \frac{z_t - tz_1}{1-t} - h_\theta(z_t, t) \right\|^2 \tag{64}$$

$$= \frac{1}{2}\frac{1}{(1-t)^2} \left\| z_1 - z_t - (1-t)h_\theta(z_t, t) \right\|^2 \tag{65}$$

$$= \frac{1}{2}\frac{1}{(1-t)^2} \left\| z_1 - \hat{h}_\theta(z_t, t) \right\|^2 \tag{66}$$

$$= \frac{\beta_t}{2} \left\| z_1 - \hat{h}_\theta(z_t, t) \right\|^2 \tag{67}$$

where $\hat{h}_\theta(z_t, t) \equiv z_t + (1-t)h_\theta(z_t, t)$ and $\beta_t \equiv 1/(1-t)^2$. In this form, $\hat{h}$ can be viewed as a denoiser.

### C.4 NoisePred

This parameterization is applicable only when $z_0$ is Gaussian. Similar to the previous section, we can construct the noise prediction parameterization by substituting $z_1$ using $z_t = tz_1 + \sqrt{(1-t)(\sigma^2 t + 1 - t)}z_0$.

$$\frac{1}{2}\left\| z_1 - \sqrt{\frac{\sigma^2 t + 1 - t}{1-t}}z_0 - h_\theta(z_t, t) \right\|^2 \tag{68}$$

$$= \frac{1}{2}\left\| \frac{z_t - \sqrt{(1-t)(\sigma^2 t + 1 - t)}z_0}{t} - \sqrt{\frac{\sigma^2 t + 1 - t}{1-t}}z_0 - h_\theta(z_t, t) \right\|^2 \tag{69}$$

$$= \frac{1}{2}\left\| \frac{\sqrt{1-t}z_t - (1-t)\sqrt{\sigma^2 t + 1 - t}z_0 - t\sqrt{\sigma^2 t + 1 - t}z_0}{t\sqrt{1-t}} - h_\theta(z_t, t) \right\|^2 \tag{70}$$

$$= \frac{1}{2}\left\| \frac{\sqrt{1-t}z_t - \sqrt{\sigma^2 t + 1 - t}z_0}{t\sqrt{1-t}} - h_\theta(z_t, t) \right\|^2 \tag{71}$$

$$= \frac{1}{2}\frac{1}{t^2(1-t)}\left\| \sqrt{1-t}z_t - \sqrt{\sigma^2 t + 1 - t}z_0 - t\sqrt{1-t}h_\theta(z_t, t) \right\|^2 \tag{72}$$

$$= \frac{1}{2}\frac{\sigma^2 t + 1 - t}{t^2(1-t)}\left\| z_0 - \frac{\sqrt{1-t}z_t - t\sqrt{1-t}h_\theta(z_t, t)}{\sqrt{\sigma^2 t + 1 - t}} \right\|^2 \tag{73}$$

$$= \frac{\beta_t}{2}\left\| z_0 - \hat{h}_\theta(z_t, t) \right\|^2 \tag{74}$$

where $\hat{h}_\theta(z_t, t) \equiv (\sqrt{1-t}z_t - t\sqrt{1-t}h_\theta(z_t, t))/\sqrt{\sigma^2 t + 1 - t}$ and $\beta_t \equiv 1/(t^2(1-t))$.

## D  LATENT SCORE FUNCTION WITH GAUSSIAN $p_0$

When $p_0(z_0)$ is gaussian, $z_0 \sim N(0, I)$, we can compute the score function estimate $\nabla_{z_t} \ln p_t(z_t)$ from the learned drift $h_\theta$ (Singh & Fischer, 2024). When $z_0$ is gaussian, the transition density $p(z_t|z_1)$ is Gaussian. With $z_t = \eta_t \epsilon + \kappa_t z_1 + \nu_t z_0$, we can reparameterize as $z_t = \kappa_t z_1 + \sqrt{\nu_t^2 + \eta_t^2}z_0, z_0 \sim N(0, I)$.

$$p(z_t|z_1) = N(z_t; \kappa_t z_1, (\nu_t^2 + \eta_t^2)I) \tag{75}$$

From Singh & Fischer (2024)(eq. 41, Appendix B) we have

$$\nabla_{z_t} \ln p_t(z_t) = \mathbb{E}_{p_t(z_1|z_t)}\left[ \frac{-z_t + \mu(z_1, t)}{\sigma(z_1, t)^2} \right] \tag{76}$$

Substituting

$$\nabla_{z_t} \ln p_t(z_t) = \mathbb{E}_{p_t(z_1|z_t)}\left[ \frac{-z_t + \kappa_t z_1}{\nu_t^2 + \eta_t^2} \right] \tag{77}$$

$$= \frac{-z_t + \kappa_t \mathbb{E}[z_1|z_t]}{\nu_t^2 + \eta_t^2} \tag{78}$$

Since the interpolation relates $z_0, z_1, z_t$ as $z_t = \kappa_t z_1 + \sqrt{\nu_t^2 + \eta_t^2}z_0$, we can rewrite the above expression in terms of $z_0$ as following

$$\nabla_{z_t} \ln p_t(z_t) = -\frac{\mathbb{E}[z_0|z_t]}{\sqrt{\nu_t^2 + \eta_t^2}} \tag{79}$$

## E  LATENT SCORE FUNCTION WITH GENERAL $p_0$

For a general distribution $p_0(z_0)$, it may not be possible to estimate the score function $\nabla_{z_t} \ln p_t(z_t)$ from the learned drift $h_\theta(z_t, t)$ alone. Here we derive the expression for estimating the score function

for a general distribution $p_0(z_0)$. Recall from eq. (10) that $p(z_t|z_0, z_1)$ is Gaussian. From Denoising Score Matching (Vincent, 2011), we can write

$$\nabla_{z_t} \ln p_t(z_t) = \mathbb{E}_{p_t(z_0, z_1|z_t)} \frac{\partial \ln p_t(z_t|z_0, z_1)}{\partial z_t} \tag{80}$$

where we have conditioned on both variables $x_0, x_1$. Since $p(z_t|z_0, z_1)$ is Gaussian, as in the previous section, we can write

$$\nabla_{z_t} \ln p_t(z_t) = \mathbb{E}_{p_t(z_0, z_1|z_t)} \left[ \frac{-z_t + \mu(z_0, z_1, t)}{\sigma(z_0, z_1, t)^2} \right] \tag{81}$$

Now, for $z_t = \eta_t \epsilon + \kappa_t z_1 + \nu_t z_0$, we have $p(z_t|z_0, z_1) = N(z_t; \kappa_t z_1 + \nu_t z_0, \eta_t^2 I)$. Substituting

$$\nabla_{z_t} \ln p_t(z_t) = \mathbb{E}_{p_t(z_0, z_1|z_t)} \left[ \frac{-z_t + \kappa_t z_1 + \nu_t z_0}{\eta_t^2} \right] \tag{82}$$

$$= \mathbb{E}_{p_t(\epsilon|z_t)} \left[ \frac{-\eta_t \epsilon}{\eta_t^2} \right] \tag{83}$$

$$= -\frac{\mathbb{E}_{p_t(\epsilon|z_t)}[\epsilon]}{\eta_t} \equiv -\frac{\mathbb{E}[\epsilon|z_t]}{\eta_t} \tag{84}$$

Note that this result mirrors the one for SI(Theorem 2.8, (Albergo et al., 2023)), though our derivation is straightforward and follows directly from Denoising Score Matching (Vincent, 2011).

## F  DETAILED DERIVATION OF SAMPLING

For an SDE of the form

$$dz_t = h_\theta(z_t, t)dt + \sigma_t dw_t \tag{85}$$

Singh & Fischer (2024) (Corollary 1) derives a flexible family of samplers as following

$$dz_t = \left[ h_\theta(z_t, t) - \frac{(1 - \gamma_t^2)\sigma_t^2}{2} \nabla_{z_t} \ln p_t(z_t) \right] dt + \gamma_t \sigma_t dw_t \tag{86}$$

where $\gamma_t$ is a time dependent weighting that can be chosen to control the amount of stochasticity injected into the sampling. Note that choosing $\gamma_t = 0$ yields the probability flow ODE (Song et al., 2020b) and results in a deterministic sampler. This general form of sampler requires both the drift $h_\theta(z_t, t)$ and the score function $\nabla_{z_t} \ln p_t(z_t)$. In general, the score function needs to be separately estimated. See section E for an estimator. We can also set $\gamma_t = 1$, leading to direct discretization of the original SDE in eq. (85). However, for the special case of Gaussian $z_0$, we can infer the score function from the learned drift $h_\theta$ (section D). For this special case, we use the general form above to derive a family of samplers for various parameterizations discussed in section C. Recall that for the choice of $\kappa_t = t, \nu_t = 1 - t$ used in this paper, the loss term is specified by eq. (50). Without any reparameterization, we have

$$h_\theta(z_t, t) = \frac{\mathbb{E}[z_1|z_t] - z_t}{1 - t} \tag{87}$$

$$\mathbb{E}[z_1|z_t] = z_t + (1 - t)h_\theta(z_t, t) \tag{88}$$

We can use the above to determine the expression for the score function

$$\nabla_x \ln p_t(z_t) = \frac{-z_t + th_\theta(z_t, t)}{\sigma^2 t + 1 - t} \tag{89}$$

Above expressions for the score $\nabla_x \ln p_t(z_t)$ can then be plugged into eq. (86) to derive a sampler for the original formulation

$$dz_t = \left[ h_\theta(z_t, t) - \frac{(1 - \gamma_t^2)\sigma^2}{2} \frac{-z_t + th_\theta(z_t, t)}{\sigma^2 t + 1 - t} \right] dt + \gamma_t \sigma dw_t \tag{90}$$

For each of the following parameterizations, we calculate the expression for the drift $h_\theta$ and the score function $\nabla_x \ln p_t(z_t)$. These expressions can then be plugged into eq. (86) to derive the sampler.

### F.1 SAMPLER FOR OrigFlow

For the OrigFlow parameterization, we have

$$h_\theta(z_t, t) = \frac{\hat{h}_\theta(z_t, t)}{\sqrt{1-t}} \tag{91}$$

For Gaussian $z_0$, we can now substitute into the expression for the score function

$$\nabla_x \ln p_t(z_t) = \frac{-z_t + th_\theta(z_t, t)}{\sigma^2 t + 1 - t} \tag{92}$$

$$= \frac{-\sqrt{1-t}z_t + t\hat{h}_\theta(z_t, t)}{\sqrt{1-t}(\sigma^2 t + 1 - t)} \tag{93}$$

The drift $h_\theta$ and the score function $\nabla_x \ln p_t(z_t)$ can now be plugged into eq. (86) to derive the sampler.

### F.2 SAMPLER FOR InterpFlow

For the InterpFlow parameterization, we have

$$h_\theta(z_t, t) = \frac{\hat{h}(z_t, t) - \sqrt{t}z_t}{\sqrt{1-t}} \tag{94}$$

For Gaussian $z_0$, we can now substitute into the expression for the score function

$$\nabla_x \ln p_t(z_t) = \frac{-z_t + th_\theta(z_t, t)}{\sigma^2 t + 1 - t} \tag{95}$$

$$= \frac{-\sqrt{1-t}z_t + t\hat{h}_\theta(z_t, t) - t\sqrt{t}z_t}{\sqrt{1-t}(\sigma^2 t + 1 - t)} \tag{96}$$

$$= \frac{-(\sqrt{1-t} + t\sqrt{t})z_t + t\hat{h}_\theta(z_t, t)}{\sqrt{1-t}(\sigma^2 t + 1 - t)} \tag{97}$$

The drift $h_\theta$ and the score function $\nabla_x \ln p_t(z_t)$ can now be plugged into eq. (86) to derive the sampler.

### F.3 SAMPLER FOR Denoising

For the Denoising parameterization, we have

$$h_\theta(z_t, t) = \frac{\hat{h}_\theta(z_t, t) - z_t}{1-t} \tag{98}$$

For Gaussian $z_0$, substituting into the expression for the score function

$$\nabla_x \ln p_t(z_t) = \frac{-z_t + th_\theta(z_t, t)}{\sigma^2 t + 1 - t} \tag{99}$$

$$= \frac{-(1-t)z_t + t\hat{h}_\theta(z_t, t) - tz_t}{(1-t)(\sigma^2 t + 1 - t)} \tag{100}$$

$$= \frac{-z_t + t\hat{h}_\theta(z_t, t)}{(1-t)(\sigma^2 t + 1 - t)} \tag{101}$$

The drift $h_\theta$ and the score function $\nabla_x \ln p_t(z_t)$ can now be plugged into eq. (86) to derive the sampler.

### F.4 SAMPLER FOR NoisePred

Again, we have

$$h_\theta(z_t, t) = \frac{-\sqrt{\sigma^2 t + 1 - t}\hat{h}_\theta(z_t, t) + \sqrt{1-t}z_t}{t\sqrt{1-t}} \tag{102}$$

For Gaussian $z_0$, substituting into the expression for the score function

$$\nabla_x \ln p_t(z_t) = \frac{-z_t + t h_\theta(z_t, t)}{\sigma^2 t + 1 - t} \tag{103}$$

$$= \frac{-\sqrt{1-t} z_t - \sqrt{\sigma^2 t + 1 - t} \hat{h}_\theta(z_t, t) + \sqrt{1-t} z_t}{\sqrt{1-t}(\sigma^2 t + 1 - t)} \tag{104}$$

$$= \frac{-\hat{h}_\theta(z_t, t)}{\sqrt{(1-t)(\sigma^2 t + 1 - t)}} \tag{105}$$

The drift $h_\theta$ and the score function $\nabla_x \ln p_t(z_t)$ can now be plugged into eq. (86) to derive the sampler.

## G   GAUSSIANITY OF CONDITIONAL DENSITY

We have

$$p(z_t|z_1, z_0) = \frac{p(z_1|z_t)p(z_t|z_0)}{p(z_1|z_0)} \tag{106}$$

Further, for the SDE in eq. (7), using results from section L, we have that the transition density $p(x_t|x_s)$ is normal with

$$p(x_t|x_s) = N(x_t; \mu_{st}, \Sigma_{st}) \tag{107}$$

$$\mu_{st} = \mu_s \exp\left(\int_s^t h(\tau)d\tau\right) \equiv \mu_s a_{st} \tag{108}$$

$$\Sigma_{st} = I \int_s^t \sigma(\tau)^2 \exp\left(2\int_\tau^t h(u)du\right) d\tau) \equiv I b_{st} \tag{109}$$

Then, the conditional density $p(z_t|z_1, z_0)$ is also normal $N(z_t; \mu(z_0, z_1, t), \Sigma(z_0, z_1, t))$ with

$$\mu(z_0, z_1, t) = \frac{b_{0t} a_{t1} z_1 + b_{t1} a_{0t} z_0}{b_{01}} \tag{110}$$

$$\Sigma(z_0, z_1, t) = \frac{b_{0t} b_{t1}}{b_{01}} I \tag{111}$$

**Proof:** First note that

$$a_{01} = a_{0t} a_{t1} \tag{112}$$

$$a_{st} = \frac{a_{0t}}{a_{0s}} = \frac{a_{s1}}{a_{t1}} \tag{113}$$

$$b_{st} = \int_s^t \sigma(v)^2 a_{vt}^2 dv \tag{114}$$

Next

$$b_{01} = \int_0^1 \sigma(v)^2 a_{v1}^2 dv \tag{115}$$

$$= \int_0^t \sigma(v)^2 a_{v1}^2 dv + \int_t^1 \sigma(v)^2 a_{v1}^2 dv \tag{116}$$

$$= \int_0^t \sigma(v)^2 a_{vt}^2 a_{t1}^2 dv + b_{t1} \tag{117}$$

$$= a_{t1}^2 \int_0^t \sigma(v)^2 a_{vt}^2 dv + b_{t1} \tag{118}$$

$$= a_{t1}^2 b_{0t} + b_{t1} \tag{119}$$

Now

$$p(z_t|z_1, z_0) = \left( \frac{1}{2\pi} \frac{b_{01}}{b_{t1}b_{0t}} \right)^{\frac{n}{2}} \exp\left( -\frac{1}{2} \left( \frac{|z_1 - a_{t1}z_t|^2}{b_{t1}} + \frac{|z_t - a_{0t}z_0|^2}{b_{0t}} - \frac{|z_1 - a_{01}z_0|^2}{b_{01}} \right) \right) \tag{120}$$

Using the identities $a_{01} = a_{0t}a_{t1}$, $b_{01} = a_{t1}^2 b_{0t} + b_{t1}$ and completing the squares we get

$$p(z_t|z_1, z_0) = \left( \frac{1}{2\pi} \frac{b_{01}}{b_{0t}b_{t1}} \right)^{\frac{n}{2}} \exp\left( -\frac{1}{2} \frac{b_{01}}{b_{0t}b_{t1}} \left| z_t - \frac{b_{0t}a_{t1}z_1 + b_{t1}a_{0t}z_0}{b_{01}} \right|^2 \right) \tag{121}$$

We can therefore parameterize $z_t$ as following using the reparameterization trick.

$$z_t = \underbrace{\sqrt{\frac{b_{0t}b_{t1}}{b_{01}}}}_{\eta_t} \epsilon + \underbrace{\frac{b_{0t}a_{t1}}{b_{01}}}_{\kappa_t} z_1 + \underbrace{\frac{b_{t1}a_{0t}}{b_{01}}}_{\nu_t} z_0, \quad \epsilon \sim N(0, I) \tag{122}$$

we can succinctly rewrite the above as

$$z_t = \eta_t \epsilon + \kappa_t z_1 + \nu_t z_0, \quad \epsilon \sim N(0, I) \tag{123}$$

Where $\eta_t, \kappa_t, \nu_t$ are appropriate scalar functions of time $t$.

## H  GENERAL TRAINING OBJECTIVE

Here we derive the form of the general training objective. The first term in the objective is the reconstruction term and remains as is. The second term of the training objective uses $u(z_t, t)$, let's recall it's expression

$$u(z_t, t) = \sigma_t^{-1}[h_t z_t + \sigma_t^2 \nabla_{z_t} \ln p(z_1|z_t) - h_\theta(z_t, t)] \tag{124}$$

The first two terms in the above serve as the target for $h_\theta$. Next, we rewrite them in terms of existing variables. Let $\xi(t)$ denote these two terms and substitute eq. (8) as following

$$\xi(t) = h_t z_t + \sigma_t^2 \nabla_{z_t} \ln p(z_1|z_t) \tag{125}$$

$$= h_t z_t + \frac{\sigma_t^2 a_{t1}(z_1 - a_{t1}z_t)}{b_{t1}} \tag{126}$$

$$= \left( h_t - \frac{\sigma_t^2 a_{t1}^2}{b_{t1}} \right) z_t + \frac{\sigma_t^2 a_{t1} z_1}{b_{t1}} \tag{127}$$

Next, recall the stochastic interpolant and the expressions for $a_{st}$ and $b_{st}$ from section G

$$z_t = \eta_t \epsilon + \kappa_t z_1 + \nu_t z_0, \quad \epsilon \sim N(0, I) \tag{128}$$

$$\eta_t = \sqrt{\frac{b_{0t}b_{t1}}{b_{01}}}, \quad \kappa_t = \frac{b_{0t}a_{t1}}{b_{01}}, \quad \nu_t = \frac{b_{t1}a_{0t}}{b_{01}}, \tag{129}$$

$$a_{st} = \exp\left( \int_s^t h(\tau)d\tau \right), \quad b_{st} = \int_s^t \sigma(v)^2 a_{vt}^2 dv \tag{130}$$

$$\tag{131}$$

Intuitively, we expect the drift $h_\theta$ to be related to the velocity field. Therefore, we compute the time derivatives of $\kappa_t, \nu_t$ and $\eta_t$ next

$$\frac{d\kappa_t}{dt} = \frac{1}{b_{01}} \left( b_{0t} \frac{da_{t1}}{dt} + \frac{db_{0t}}{dt} a_{t1} \right) \tag{132}$$

$$\frac{d\nu_t}{dt} = \frac{1}{b_{01}} \left( b_{t1} \frac{da_{0t}}{dt} + \frac{db_{t1}}{dt} a_{0t} \right) \tag{133}$$

$$\frac{d\eta_t}{dt} = \frac{1}{2\eta_t b_{01}} \left( b_{0t} \frac{db_{t1}}{dt} + \frac{db_{0t}}{dt} b_{t1} \right) \tag{134}$$

From the expression for $a_{st}$, using differentiation under the integral sign, we have

$$\frac{da_{0t}}{dt} = a_{0t}h_t, \quad \frac{da_{t1}}{dt} = -a_{t1}h_t \tag{135}$$

Similarly, from the expression for $b_{st}$

$$\frac{db_{0t}}{dt} = \sigma_t^2 a_{tt}^2 + 2\int_0^t \sigma(v)^2 a_{vt}^2 h_t dv = \sigma_t^2 + 2b_{0t}h_t \tag{136}$$

$$\frac{db_{t1}}{dt} = -\sigma_t^2 a_{t1}^2 \tag{137}$$

Since $a_{tt} = 1$. Substituting back into the equations for the derivatives of $\kappa_t$ and $\nu_t$

$$\frac{d\kappa_t}{dt} = \frac{1}{b_{01}}\left(-b_{0t}a_{t1}h_t + (\sigma_t^2 + 2b_{0t}h_t)a_{t1}\right) = \frac{1}{b_{01}}\left(\sigma_t^2 a_{t1} + b_{0t}a_{t1}h_t\right) \tag{138}$$

$$= \frac{\sigma_t^2 a_{t1}}{b_{01}} + \kappa_t h_t \tag{139}$$

$$\frac{d\nu_t}{dt} = \frac{1}{b_{01}}\left(b_{t1}a_{0t}h_t - \sigma_t^2 a_{t1}^2 a_{0t}\right) = \nu_t h_t - \frac{\sigma_t^2 a_{t1}^2 a_{0t}}{b_{01}} \tag{140}$$

$$= \nu_t\left(h_t - \frac{\sigma_t^2 a_{t1}^2}{b_{t1}}\right) \tag{141}$$

$$\frac{d\eta_t}{dt} = \frac{1}{2\eta_t b_{01}}\left(-\sigma_t^2 a_{t1}^2 b_{0t} + (\sigma_t^2 + 2b_{0t}h_t)b_{t1}\right) \tag{142}$$

$$= \frac{1}{2\eta_t b_{01}}\left((b_{t1} - a_{t1}^2 b_{0t})\sigma_t^2 + 2b_{0t}b_{t1}h_t\right) \tag{143}$$

$$= \frac{1}{2\eta_t b_{01}}\left((b_{t1} - a_{t1}^2 b_{0t})\sigma_t^2 + 2b_{0t}\left(\frac{b_{t1}}{\nu_t}\frac{d\nu_t}{dt} + \sigma_t^2 a_{t1}^2\right)\right) \tag{144}$$

$$= \frac{1}{2\eta_t b_{01}}\left((b_{t1} + a_{t1}^2 b_{0t})\sigma_t^2 + \frac{2b_{0t}b_{t1}}{\nu_t}\frac{d\nu_t}{dt}\right) \tag{145}$$

$$= \frac{1}{2\eta_t b_{01}}\left(b_{01}\sigma_t^2 + \frac{2\eta_t^2 b_{01}}{\nu_t}\frac{d\nu_t}{dt}\right) \tag{146}$$

$$= \frac{\sigma_t^2}{2\eta_t} + \frac{\eta_t}{\nu_t}\frac{d\nu_t}{dt} \tag{147}$$

Where we have used the identity $b_{01} = b_{t1} + a_{t1}^2 b_{0t}$ from eq. (119). Further, we can relate $\frac{d\kappa_t}{dt}$ and $\frac{d\nu_t}{dt}$ by eliminating $h_t$ as following

$$\frac{d\kappa_t}{dt} = \frac{\sigma_t^2 a_{t1}}{b_{01}} + \kappa_t\left(\frac{1}{\nu_t}\frac{d\nu_t}{dt} + \frac{\sigma_t^2 a_{t1}^2}{b_{t1}}\right) = \frac{\kappa_t}{\nu_t}\frac{d\nu_t}{dt} + \frac{\sigma_t^2 a_{t1}}{b_{01}} + \kappa_t\frac{\sigma_t^2 a_{t1}^2}{b_{t1}} \tag{148}$$

$$= \frac{\kappa_t}{\nu_t}\frac{d\nu_t}{dt} + \frac{\sigma_t^2 a_{t1}}{b_{01}} + \frac{b_{0t}a_{t1}}{b_{01}}\frac{\sigma_t^2 a_{t1}^2}{b_{t1}} = \frac{\kappa_t}{\nu_t}\frac{d\nu_t}{dt} + \frac{\sigma_t^2 a_{t1}(b_{t1} + b_{0t}a_{t1}^2)}{b_{01}b_{t1}} \tag{149}$$

$$= \frac{\kappa_t}{\nu_t}\frac{d\nu_t}{dt} + \frac{\sigma_t^2 a_{t1}b_{01}}{b_{01}b_{t1}} \tag{150}$$

$$= \frac{\kappa_t}{\nu_t}\frac{d\nu_t}{dt} + \frac{\sigma_t^2 a_{t1}}{b_{t1}} \tag{151}$$

We can now substitute into the expression for $\xi(t)$ in eq. (127)

$$\xi(t) = \frac{1}{\nu_t}\frac{d\nu_t}{dt}z_t + \frac{\sigma_t^2 a_{t1}z_1}{b_{t1}} \tag{152}$$

$$= \frac{1}{\nu_t}\frac{d\nu_t}{dt}(\eta_t\epsilon + \kappa_t z_1 + \nu_t z_0) + \left(\frac{d\kappa_t}{dt} - \frac{\kappa_t}{\nu_t}\frac{d\nu_t}{dt}\right)z_1 \tag{153}$$

$$= \frac{\eta_t}{\nu_t}\frac{d\nu_t}{dt}\epsilon + \frac{d\kappa_t}{dt}z_1 + \frac{d\nu_t}{dt}z_0 \tag{154}$$

$$= \left(\frac{d\eta_t}{dt} - \frac{\sigma_t^2}{2\eta_t}\right)\epsilon + \frac{d\kappa_t}{dt}z_1 + \frac{d\nu_t}{dt}z_0 \tag{155}$$

Substituting back into the expression for $u(z_t, t)$ we can write the general form as following

$$u(z_t, t) = \sigma_t^{-1} \left[ \left( \frac{d\eta_t}{dt} - \frac{\sigma_t^2}{2\eta_t} \right) \epsilon + \frac{d\kappa_t}{dt} z_1 + \frac{d\nu_t}{dt} z_0 - h_\theta(z_t, t) \right] \tag{156}$$

With the $u(z_t, t)$ above, the ELBO can be written using eq. (3).

## I    DRIFT $h_t$, DISPERSION $\sigma_t$ AND STOCHASTICITY $\eta_t$ FROM $\kappa_t, \nu_t$

Often, specifying the interpolant coefficients $\kappa_t, \nu_t$ is intuitively easier than specifying $h_t, \sigma_t$ directly. Here we derive expressions for $h_t$ and $\sigma_t$ given $\kappa_t$ and $\nu_t$. We have

$$\frac{d\kappa_t}{dt} = \kappa_t h_t + \frac{\sigma_t^2 a_{t1}}{b_{01}} \tag{157}$$

$$\frac{d\nu_t}{dt} = h_t \nu_t - \frac{\sigma_t^2 a_{t1}^2}{b_{t1}} \nu_t \tag{158}$$

Multiplying first equation by $\nu_t$ and second by $\kappa_t$ and then subtracting the second from the first

$$\nu_t \frac{d\kappa_t}{dt} - \kappa_t \frac{d\nu_t}{dt} = \nu_t \frac{\sigma_t^2 a_{t1}}{b_{01}} + \kappa_t \frac{\sigma_t^2 a_{t1}^2}{b_{t1}} \nu_t \tag{159}$$

$$= \left( \nu_t \frac{\sigma_t^2 a_{t1}}{b_{01}} + \kappa_t \frac{\sigma_t^2 a_{t1}^2}{b_{t1}} \nu_t \right) \tag{160}$$

Substituting in the definitions of $\kappa_t$ and $\nu_t$ in RHS and simplifying

$$\nu_t \frac{d\kappa_t}{dt} - \kappa_t \frac{d\nu_t}{dt} = \left( \frac{b_{t1} a_{01} \sigma_t^2}{b_{01}^2} + \frac{b_{0t} \sigma_t^2 a_{t1}^2 a_{01}}{b_{01}^2} \right) \tag{161}$$

$$= \frac{a_{01} \sigma_t^2}{b_{01}^2} \left( b_{t1} + b_{0t} a_{t1}^2 \right) = \frac{a_{01} \sigma_t^2}{b_{01}^2} b_{01} \tag{162}$$

$$= \frac{a_{01} \sigma_t^2}{b_{01}} \tag{163}$$

where we have used $a_{01} = a_{0t} a_{t1}$ and $b_{01} = b_{t1} + b_{0t} a_{t1}^2$. Therefore

$$\sigma_t^2 = \frac{b_{01}}{a_{01}} \left( \nu_t \frac{d\kappa_t}{dt} - \kappa_t \frac{d\nu_t}{dt} \right) \tag{164}$$

Where $b_{01} > 0, a_{01} > 0$ are time $t$ independent constants that can't be determined by $\kappa_t, \nu_t$ alone. In this paper, we assume $a_{01} = 2$ and $b_{01} = a_{01} \sigma^2$, where $\sigma$ is a hyper-parameter. Next, to derive the expression for $h_t$, we eliminate $\sigma_t^2$ from eqs. (157) and (158).

$$b_{01} \left( \frac{d\kappa_t}{dt} - \kappa_t h_t \right) = \frac{b_{t1}}{a_{t1}} \left( -\frac{1}{\nu_t} \frac{d\nu_t}{dt} + h_t \right) \tag{165}$$

$$h_t \left( b_{01} \kappa_t + \frac{b_{t1}}{a_{t1}} \right) = b_{01} \frac{d\kappa_t}{dt} + \frac{b_{t1}}{a_{t1}} \frac{1}{\nu_t} \frac{d\nu_t}{dt} \tag{166}$$

$$h_t \left( \frac{a_{t1} b_{01} \kappa_t + b_{t1}}{a_{t1}} \right) = b_{01} \frac{d\kappa_t}{dt} + \frac{b_{t1}}{a_{t1}} \frac{b_{01}}{b_{t1} a_{0t}} \frac{d\nu_t}{dt} \tag{167}$$

$$h_t \left( a_{0t} a_{t1} \kappa_t + \frac{a_{0t} b_{t1}}{b_{01}} \right) = a_{0t} a_{t1} \frac{d\kappa_t}{dt} + \frac{d\nu_t}{dt} \tag{168}$$

$$h_t \left( a_{01} \kappa_t + \nu_t \right) = a_{01} \frac{d\kappa_t}{dt} + \frac{d\nu_t}{dt} \tag{169}$$

$$h_t = \frac{a_{01} \frac{d\kappa_t}{dt} + \frac{d\nu_t}{dt}}{a_{01} \kappa_t + \nu_t} \tag{170}$$

As before, $a_{01} > 0$ is a time independent constant that can't be determined from the choice of $\kappa_t, \nu_t$ alone. Finally, to express $\eta_t$ in terms of given $\kappa_t, \nu_t$, note that

$$\eta_t^2 = \frac{b_{0t} b_{t1}}{b_{01}} = \frac{b_{01}}{a_{0t} a_{t1}} \frac{b_{0t} a_{t1}}{b_{01}} \frac{b_{t1} a_{0t}}{b_{01}} = \frac{b_{01}}{a_{01}} \kappa_t \nu_t \tag{171}$$

where we have used the identity $a_{01} = a_{0t}a_{t1}$. In the following, we derive the formulation for the linear $\kappa_t, \nu_t$ schedule used in experiments in this paper. This schedule also corresponds to the choice used in Stochastic Interpolants(Albergo et al., 2023). Note that similar choice is made by the Rectified Flow (Liu et al., 2022), however the missing $\eta$ term implies that they do not have a bound on the likelihood, as also observed by Albergo et al. (2023). We also provide the derivation for the variance preserving schedule as it is quite commonly used for diffusion models. However, we do not empirically explore it.

## J  FORMULATION FOR LINEAR $\kappa_t, \nu_t$

For linear choice $\kappa_t = t, \nu_t = 1 - t$. Further, we assume $a_{01} = 2, b_{01} = a_{01}\sigma^2$. Therefore,

$$\frac{d\kappa_t}{dt} = 1, \quad \frac{d\nu_t}{dt} = -1 \tag{172}$$

We can write the expressions for $h_t$ and $\sigma_t^2$ directly, using eqs. (164) and (170), as

$$h_t = \frac{1}{1+t}, \quad \sigma_t^2 = \sigma^2 \tag{173}$$

To express the latent stochastic interpolant, we can calculate the coefficient $\eta_t$ for $\epsilon$

$$\eta_t = \sqrt{\frac{b_{01}}{a_{01}}\kappa_t\nu_t} = \sigma\sqrt{t(1-t)} \tag{174}$$

We can now write the expression for the latent stochastic interpolant

$$z_t = \sigma\sqrt{t(1-t)}\epsilon + tz_1 + (1-t)z_0, \quad \epsilon \sim N(0, I). \tag{175}$$

Finally, to express $u(z_t, t)$ first we calculate

$$\frac{d\eta_t}{dt} - \frac{\sigma_t^2}{2\eta_t} = \frac{\sigma(1-t-t)}{2\sqrt{t(1-t)}} - \frac{\sigma^2}{2\sigma\sqrt{t(1-t)}} = \frac{\sigma^2(1-2t) - \sigma^2}{2\sigma\sqrt{t(1-t)}} = -\sigma\sqrt{\frac{t}{1-t}} \tag{176}$$

leading to

$$u(z_t, t) = \sigma^{-1}\left[-\sigma\sqrt{\frac{t}{1-t}}\epsilon + z_1 - z_0 - h_\theta(z_t, t)\right] \tag{177}$$

## K  FORMULATION FOR VARIANCE PRESERVING $\kappa_t, \nu_t$

For the variance preserving formulation, we set $\kappa_t = \sqrt{t}$ and $\eta_t^2 + \nu_t^2 = 1 - t$. Note that if $z_0 \sim N(0, I)$ is Gaussian, this setting leads to the latent stochastic interpolant $z_t = \sqrt{t}z_1 + \sqrt{1-t}z_0$. Here $\epsilon$ and $z_0$ have been combined since they both are Gaussian. Let $b_{01}/a_{01} = C$, then

$$\eta_t^2 = C\sqrt{t}\nu_t = 1 - t - \nu_t^2 \tag{178}$$

$$\implies \nu_t = \frac{-C\sqrt{t} + \sqrt{(C^2 - 4)t + 4}}{2} \tag{179}$$

Using above, the expressions for $h_t$ and $\sigma_t^2$ can be derived as

$$h_t = \frac{\frac{a_{01}}{\sqrt{t}} - \frac{C}{2\sqrt{t}} + \frac{C^2 - 4}{2\sqrt{(C^2-4)t+4}}}{2a_{01}\sqrt{t} - C\sqrt{t} + \sqrt{(C^2 - 4)t + 4}} \tag{180}$$

$$\sigma_t^2 = \frac{C}{\sqrt{t}\sqrt{(C^2 - 4)t + 4}} \tag{181}$$

Choosing $a_{01} = 1$ and $C = 2$ yields

$$h_t = 0, \quad \sigma_t^2 = \frac{1}{\sqrt{t}}, \quad \nu_t = 1 - \sqrt{t} \tag{182}$$

The coefficient $\eta_t$ for $\epsilon$ can be calculated as

$$\eta_t = \sqrt{\frac{b_{01}}{a_{01}}\kappa_t\nu_t} = \sqrt{2\sqrt{t}(1-\sqrt{t})} \tag{183}$$

We can now write the expression for the latent stochastic interpolant

$$z_t = \sqrt{2\sqrt{t}(1-\sqrt{t})}\epsilon + \sqrt{t}z_1 + (1-\sqrt{t})z_0, \quad \epsilon \sim N(0,I). \tag{184}$$

Finally, to express $u(z_t, t)$ first we calculate

$$\frac{d\eta_t}{dt} - \frac{\sigma_t^2}{2\eta_t} = -\frac{1}{\sqrt{2\sqrt{t}(1-\sqrt{t})}} \tag{185}$$

with

$$\frac{d\kappa_t}{dt} = \frac{1}{2\sqrt{t}}, \quad \frac{d\nu_t}{dt} = -\frac{1}{2\sqrt{t}} \tag{186}$$

we arrive at

$$u(z_t, t) = \sigma^{-1}\left[-\frac{1}{\sqrt{2\sqrt{t}(1-\sqrt{t})}}\epsilon + \frac{1}{2\sqrt{t}}z_1 - \frac{1}{2\sqrt{t}}z_0 - h_\theta(z_t, t)\right] \tag{187}$$

Note that above expression is for a particular choice of $a_{01} = 1$ and the ratio $b_{01}/a_{01} = 2$, which we chose for relative simplicity of the final expression above. Other choices can be made, leading to different expressions.

## L   GAUSSIAN TRANSITION DENSITIES

Let's consider a linear SDE of the form

$$dz_t = h_t z_t dt + u_t dt + \sigma_t dw_t \tag{188}$$

When the SDE is linear with additive noise, we know that the transition densities are gaussian and are therefore fully specified by their mean and covariance. From Särkkä & Solin (2019) (Eq 6.2) these are specified by the following differential equations

$$\frac{d\mu_t}{dt} = h_t\mu_t + u_t \tag{189}$$

$$\frac{d\Sigma_t}{dt} = 2h_t\Sigma_t + \sigma_t^2 I \tag{190}$$

The solution to these is given by (eq. 6.3, 6.4, Särkkä & Solin (2019))

$$\mu_t = \Psi(t, t_0)\mu_{t_0} + \int_{t_0}^t \Psi(t, \tau)u(\tau)d\tau \tag{191}$$

$$\Sigma_t = \Psi(t, t_0)\Sigma_{t_0}\Psi(t, t_0)^T + \int_{t_0}^t \sigma(\tau)^2\Psi(t, \tau)\Psi(t, \tau)^T d\tau \tag{192}$$

Where $\Psi(s, t)$ is the transition matrix. For our specific case of linear SDEs, we have

$$\Psi(s, t) = \exp\left(\int_t^s h(\tau)d\tau\right) \tag{193}$$

Substituting, we get

$$\mu_t = \mu_{t_0}\exp\left(\int_{t_0}^t h(\tau)d\tau\right) + \int_{t_0}^t \exp\left(\int_\tau^t h(s)ds\right)u(\tau)d\tau \tag{194}$$

$$\Sigma_t = \Sigma_{t_0}\exp\left(2\int_{t_0}^t h(\tau)d\tau\right) + I\int_{t_0}^t \sigma(\tau)^2\exp\left(2\int_\tau^t h(s)ds\right)d\tau \tag{195}$$

## M  GAUSSIAN $z_0$

For the interpolant (section J)

$$z_t = \sigma\sqrt{t(1-t)}\epsilon + tz_1 + (1-t)z_0, \quad \epsilon \sim N(0, I), \tag{196}$$

if $z_0$ is gaussian, we can replace the linear combination of two normal random variables $\epsilon, z_0$ with a single random variable $\hat{z}_0 \sim N(\hat{\mu}, \hat{\Sigma})$. Assuming $z_0 \sim N(0, I)$, the mean $\hat{\mu} = 0$ and covariance $\hat{\Sigma}$ can be computed as

$$\hat{\Sigma} = \left(\sigma^2 t(1-t) + (1-t)^2\right) I \tag{197}$$
$$= (1-t)(t\sigma^2 + (1-t))I \tag{198}$$

Using the reparameterization trick, we can express $\hat{z}_0$ in terms of $z_0$ and write

$$z_t = tz_1 + \sqrt{(1-t)(t\sigma^2 + (1-t))}z_0, \quad z_0 \sim N(0, I) \tag{199}$$

Note that

$$z_t = tz_1 + \sqrt{1-t}z_0, \qquad \text{if } \sigma^2 = 1 \tag{200}$$
$$z_t = tz_1 + (1-t)z_0, \qquad \text{if } \sigma^2 = 0 \tag{201}$$

Similarly, recall the expression for $u(z_t, t)$ from section J

$$u(z_t, t) = \sigma^{-1}\left[-\sigma\sqrt{\frac{t}{1-t}}\epsilon + z_1 - z_0 - h_\theta(z_t, t)\right] \tag{202}$$

If $z_0 \sim N(0, I)$ is also gaussian, we can combine $\epsilon, z_0$ and write

$$u(z_t, t) = \sigma^{-1}\left[z_1 - \sqrt{\frac{1 + (\sigma^2 - 1)t}{1-t}}z_0 - h_\theta(z_t, t)\right] \tag{203}$$

if we choose $\sigma^2 = 1$, then the expression simplifies to

$$u(z_t, t) = z_1 - \frac{1}{\sqrt{1-t}}z_0 - h_\theta(z_t, t) \tag{204}$$

Finally, we would like to reiterate that we arrive at the above by assuming $z_0$ is gaussian. The general form derived in other sections make no assumptions about the distribution of $z_0$.

## N  CHOICE OF PRIOR $p_0$

The Gaussian distribution, along with a small set of other distributions, enjoys the special privilege of being Lévy stable. That is, a linear combination of two Gaussian random variables is still a Gaussian random variable. Lévy stability is the main property behind the original formulation of the simulation free training of the Gaussian diffusion models, e.g. as in DDPM. In contrast, Laplacian, Uniform and Gaussian Mixture are not Lévy stable, and thus our experiment with those provides strong evidence for the general nature of the proposed method. The Gaussian mixture used in our experiment was constructed by having a component for each training image. Consequently, it is a mixture with a very large number of components. The current estimate of the encoder being learned was used to encode the training images, yielding the means of the corresponding components. Standard deviation for each dimension was fixed to $0.1$. In practice, we simply shuffled the encoding of the training images, added noise, and used a `stop_gradient` operation to prevent the flow of gradient through the prior. Since the encoder is also evolving during training, this experiment required $\sim 3\times$ more steps to yield the reported FID. Without `stop_gradient`, the experiment became unstable.

## O  IMAGENET TRAINING AND EVALUATION DETAILS

We trained our models using the entire ImageNet training dataset, consisting of approximately 1.2 million images. Models are trained with Stochastic Gradient Descent (SGD) with the AdamW

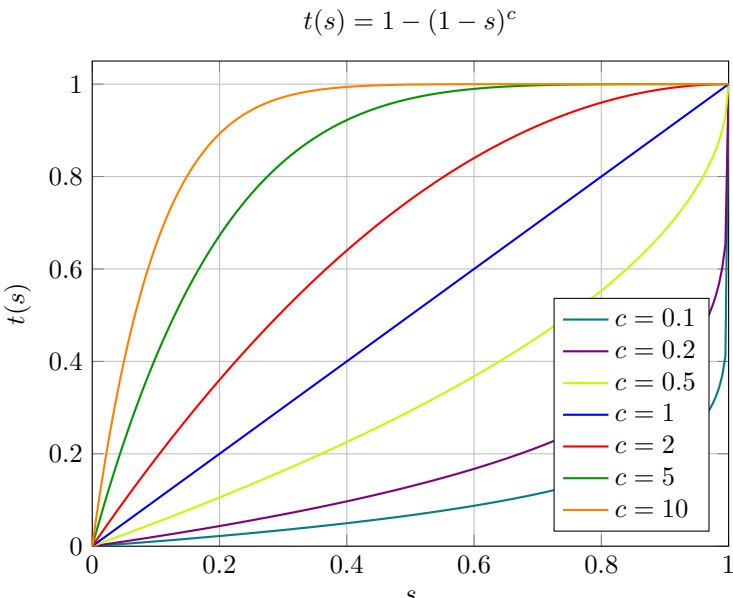

Figure 4: **Schedule for $t$.** A visualization of the schedule for $t(s)$ with $s \in [0, 1]$ as $c$ is varied. As $c$ increases, larger $t$ values are favored, thereby sampling interpolants closer to $t = 1$ more frequently.

.

optimizer (Kingma & Ba, 2014; Loshchilov & Hutter, 2017), using $\beta_1 = 0.9, \beta_2 = 0.99, \epsilon = 10^{-12}$. All models are trained for 1000 epochs using a batch size of 2048, except for the ones reported in table 1 where they were trained for 2000 epochs. Only center crops were used after resizing the images to the have the smaller side match the target resolution. For data augmentation, only horizontal (left-right) flips were used. Pixel values for an image $I$ were scaled to the range $[-1, 1]$ by computing $2(I/255) - 1$ before feeding to the model. For evaluation, a exponential moving average of the model's parameters was used using a decay rate of 0.9999. The FIDs were computed over the training dataset, with reference statistics derived from center-cropped images, without any further augmentation. All FIDs are reported with class conditioned samples. To compute PSNR, sampled image pixel values were scaled back to the range $[0, 255]$ and quantized to integer values. Figure 4 visualizes the change of variables discussed in section 4. All reported results use $c = 1$, resulting in uniform schedule, for both training and sampling, except for NoisePred and Denoising both of which resulted in slightly better FID values for $c = 2$ during sampling.

Each model was trained on Google Cloud TPU v3 with $8 \times 8$ configuration. For 2000 epochs, the $64 \times 64$ model took 2 days to train, $128 \times 128$ took 4 days to train and $256 \times 256$ took 7 days to train. For 1000 epochs, the training times were roughly the half of that for 2000 epochs. The training times for the models reported in table 1 are roughly similar for similarly sized models. Note that our training setup is not maximally optimized for training throughput.

## P ARCHITECTURE DETAILS

The base architecture of our model is adapted from the work described by Hoogeboom et al. (2023) and modified to separate out Encoder, Decoder and Latent SI models. In the adapted base architecture feature maps are processed using groups of convolution blocks and downsampled spatially after each group, to yield the lowest feature map resolution at $16 \times 16$. A sequence of Self-Attention Transformer blocks then operates on the $16 \times 16$ feature map. Note that the transformer blocks in our adapted architecture operate only at $16 \times 16$ resolution. Consequently, for a $64 \times 64$ resolution input image, two downsamplings are performed, for $128 \times 128$ resolution, three downsamplings are performed and for $256 \times 256$ four downsamplings are performed. All convolutional groups have the same number of convolutional blocks. The observation space SI models used in this paper are constructed using this adapted base architecture. To construct Encoder, Decoder and Latent SI

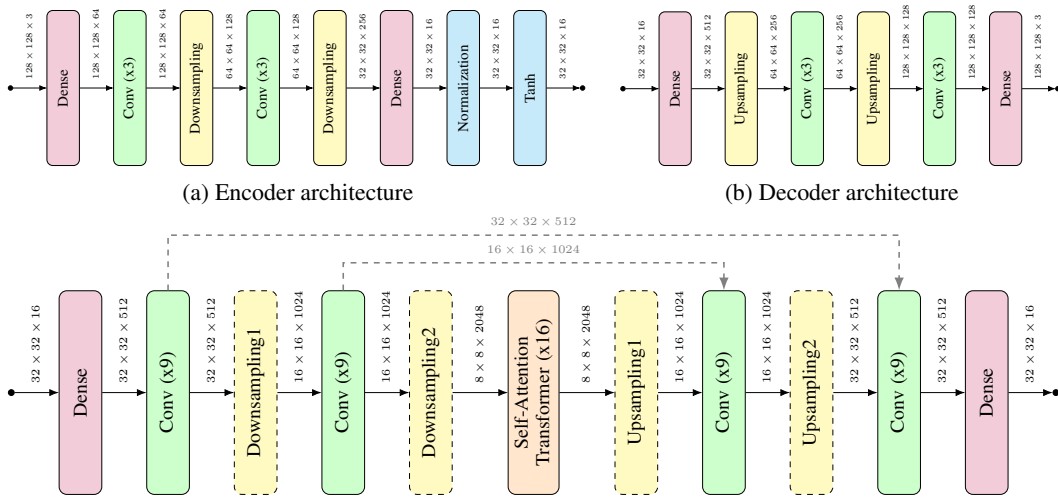

(a) Encoder architecture

(b) Decoder architecture

(c) Latent stochastic interpolant model architecture. The blocks shown with dashed boundaries are optional across different resolutions.

Figure 5: An overview of the architecture of various components for $128 \times 128$ resolution model. The architecture for $64 \times 64$ and $256 \times 256$ resolutions is similar, except for the difference in the spatial feature map sizes. See section P for details.

models, we simply partition the base model into three parts. The first part contains two groups of convolutional blocks, each followed by downsampling, and forms the encoder. An extra dense layer is added to reduce the number of channels. Further, the output is normalized to have zero mean and unit standard deviation followed by tanh activation to limit the range to $[-1, 1]$. Similarly, the last part contains two groups of convolutional blocks, each followed by upsampling, and forms the decoder. An extra dense layer is added at the beginning to increase the number of channels. The remaining middle portion forms the Latent SI model, where two extra dense layers are added, one at beginning and one at end to increase and decrease the feature map sizes respectively. We show an overview of the architecture for various components in the fig. 5.

Note that the `tanh` activation or other forms of scale control, such as normalization, play a crucial role in preventing the encoder from learning arbitrarily large embeddings and allowing it to achieve better FID. Without this constraint, the model makes the encoder outputs have large scale to make denoising easier at later timesteps. This is an important implementation detail that ensures stable training. Empirically, encoder output normalization yielded more stable training and better FID, than without anything, at the same number of steps. Addition of `tanh` further improved the FID.

For different resolutions, the Encoder and Decoder models are fully convolutional and have the same architecture. The architecture of Latent SI models differs in the presence/absence of the optional downsampling and upsampling blocks (shown as blocks with dashed boundaries). The $64 \times 64$ Latent SI model does not contain any downsampling/upsampling blocks as the encoder output is already $16 \times 16$. The $128 \times 128$ model does not contain "Downsampling1" and "Upsampling2" blocks. The $256 \times 256$ model contains all blocks. All models contain 16 Self-Attention Transformer blocks. To increase/decrease number of parameters to match model capacities, only the number of convolutional blocks in groups immediately before and after the Self-Attention Transformer blocks is changed.

All models operate with a $3\times$ smaller latent dimensionality that the observations. We focused on this dimensionality ratio to ensure fair comparison with observation-space baselines while maintaining reasonable latent dimensionality for effective modeling. In earlier experiments we tried other compression ratios including $2\times$ and $4\times$, before settling on $3\times$. The primary effect of the dimensionality ratio is on the reconstruction performance. Higher the dimensionality ratio, the harder it is for the decoder to achieve a high PSNR at the same number of training steps, resulting in worse sample quality (FID) and longer training times. Lower the dimensionality ratio, less the computational advantage.

Table 5: Comparison with state-of-the-art FID results on ImageNet $128 \times 128$. Note that these models have differing sizes, FLOPs and NFEs. The comparison is provided purely for reference.

| Method | FID |
|---|---|
| Ours | 3.12 |
| SiD2 (Hoogeboom et al., 2024) | 1.26 |
| PaGoDA (Kim et al., 2024) | 1.48 |
| DisCo-Diff (Xu et al., 2024) | 1.73 |
| VDM++ (Kingma & Gao, 2023) | 1.75 |
| SiD (Hoogeboom et al., 2023) | 1.94 |
| RIN (Jabri et al., 2022) | 2.75 |
| CDM (Ho & Salimans, 2022) | 3.52 |
| ADM (Dhariwal & Nichol, 2021) | 5.91 |

## Q  ADDITIONAL SAMPLING DETAILS AND RESULTS

All the results reported in the paper use the deterministic sampler with 300 steps, setting $\gamma_t = 0$ in eq. (86), except when otherwise stated. fig. 3 and fig. 6 use stochastic sampling with $\gamma_t \equiv \gamma(1 - t)$, where $\gamma$ is a specified constant. We use Euler (for probability flow ODE) and Euler-Maruyama (for SDE) discretization for all results, except for qualitative inversion results in fig. 3 and fig. 6. For the inversion results we experimented with two reversible samplers: 1) Reversible Heun (Kidger et al., 2021) and, 2) Asynchronous Leapfrog Integrator (Zhuang et al., 2021). While both exhibited instability and failed to invert some of the images, we found Asynchronous Leapfrog Integrator to be more stable in our experiments and used it for results in fig. 3 and fig. 6. Figure 7 provides additional samples for qualitative assessment, complementing fig. 2 in the main paper.

Sampling speed (with 100 steps) for pixel space models is roughly 2.2 images/sec/core for 64x64, 0.95 images/sec/core for 128x128 and 0.21 images/sec/core for 256x256. LSI achieves 2.65 images/sec/core for 64x64, 1.30 images/sec/core, and 0.53 images/sec/core for 256x256. We would like to emphasize that these numbers exhibit high variance, are highly hardware dependent and can be significantly impacted by hardware specific optimizations that are not the focus of this paper.

## R  COMPARISON WITH OTHER METHODS

While the primary focus of this paper is on the theoretical results and their empirical validation, in table 5 we present comparison with other image generation methods for completeness. We provide this table purely for reference as these methods are not directly comparable due to differing model sizes, FLOPs and NFEs. While our best result is comparable, techniques in these works are complementary to our method. We leave it as future work to explore this direction.

## S  USE OF LLM

LLMs were used to help create some of the figures in the paper.

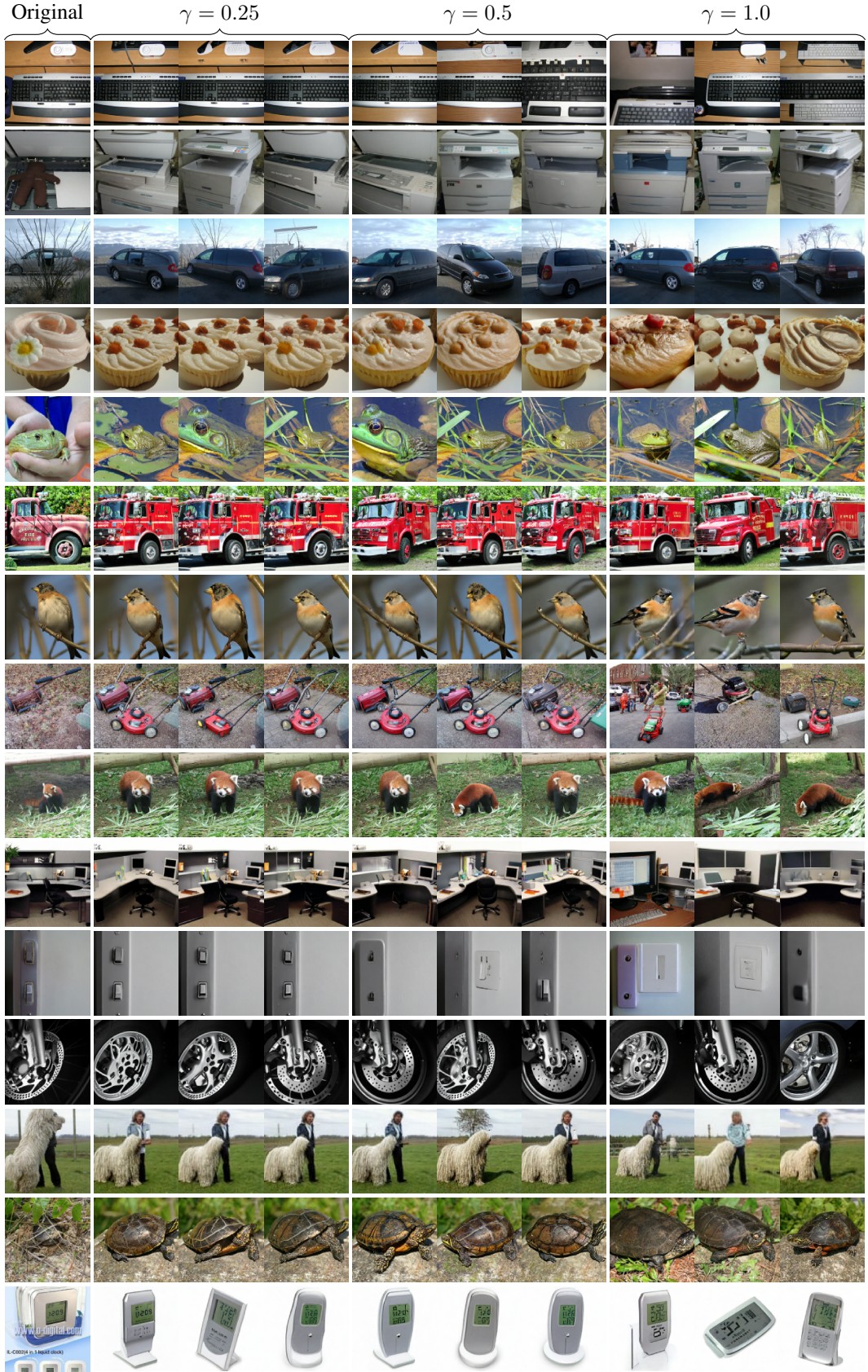

Figure 6: LSI supports flexible sampling.

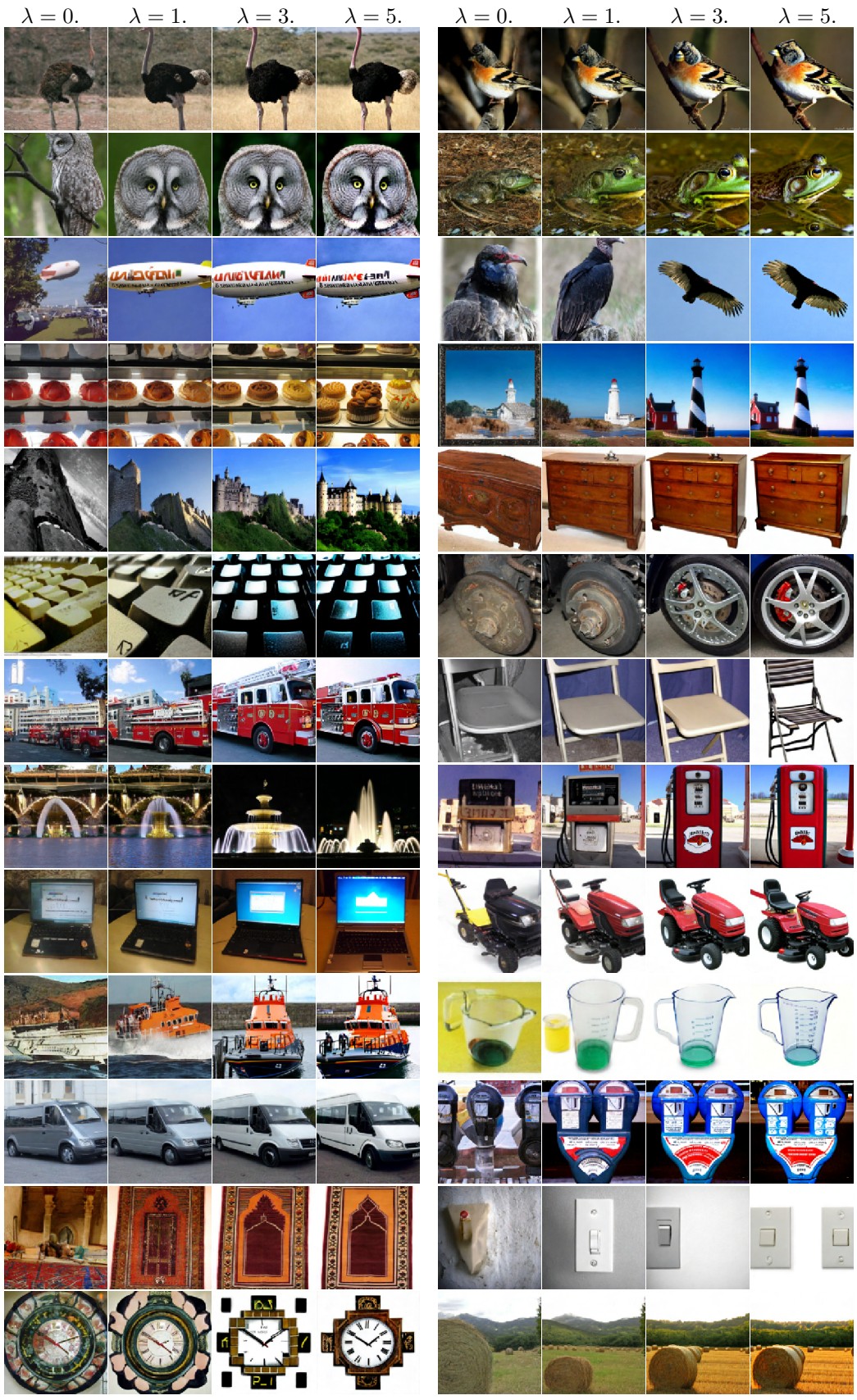

Figure 7: LSI supports CFG sampling.

