# OpenReview forum: "Latent Stochastic Interpolants"
_ICLR.cc/2026/Conference — ICLR 2026 Poster_

### Official Review · Reviewer_xrNR · 2025-10-25

**Soundness:** 3
**Presentation:** 3
**Contribution:** 3
**Rating:** 6
**Confidence:** 2

**Summary:**

The key idea of this paper is to jointly train an encoder, decoder, and latent stochastic interpolant model using a continuous-time ELBO. Unlike traditional SI, which requires direct access to samples from both prior and target distributions, LSI constructs interpolants in the latent space, allowing end-to-end optimization.

**Strengths:**

1. The paper provides generalization of stochastic interpolants into latent spaces, enabling joint encoder–decoder–generator learning.

2. The derivation of a continuous-time ELBO in latent space seems theoretically grounded.

3. Demonstrates comparable or better FIDs across multiple resolutions on ImageNet

**Weaknesses:**

The assumptions of linear SDEs and Gaussian posteriors may limit its expressivity. It’s unclear how much these approximations affect performance or generalization.

**Questions:**

This paper is well motivated and well presented. I do not have many questions, but I am curious about whether the method could be extended to a learnable prior, also formulated within the joint learning scheme? The proposed method mentioned "arbitrary prior", which, however, are mainly simple, known distributions (e.g., Gaussian, Laplace). Would you also consider a learnable prior (e.g., an EBM prior or another more sophisticated prior)?

---

> ### Author Response · Authors · 2025-11-19
> **Author Response**
>
> We thank the reviewer for the positive assessment of our work as "well motivated and well presented" and for recognizing the theoretical grounding of our continuous-time ELBO derivation.
> We have spent significant effort in making sure that our theoretical results are not only
> rigorously proved but also are presented in an accessible manner.
> We are surprised by a near borderline rating given that there are no major concerns and hope that our responses would further help clarify any remaining doubts.
>
> ### Learnable Priors within Joint Learning Framework
>
> This is an excellent question! The short answer is - Yes. However, note the following:
>
> **Current support for diverse priors**: Our method already supports "arbitrary prior" distributions as claimed and empirically demonstrated. The Gaussian Mixture experiment in **Table 4** already demonstrates this, where we evaluate LSI with a *very large* Gaussian Mixture prior where there is a separate mixture component for _each training image augmentation_. Given the universal approximation property of Gaussian mixtures, we consider
> this experiment to be a conclusive demonstration of the ability to use arbitrary priors.
> See Section L in appendix for additional details and discussion. Part of this
> explanation was moved to appendix due to lack of space. We will improve the main text to make this clear.
>
> **Theoretical compatibility with learnable priors**: Our theoretical framework (Sections 2-5) is fully compatible with learnable priors. The ELBO derivation in Equation 3 and the latent stochastic interpolant construction in Section 3 do not restrict $p_0(z_0)$ to be fixed. The key requirements are:
> 1. Ability to sample from $p_0(z_0)$
> 2. Constructing a diffusion bridge between $p_0$ and the aggregated posterior
>
> **Practical considerations**: Implementing learnable priors like EBMs within joint training would require:
> - Parameterizing $p_0(z_0)$ with additional neural networks
> - Ensuring stable sampling from the learned prior during training
> - Potentially more complex optimization dynamics
>
> We consider this to be an exciting direction for future work.
>
> ### Expressivity of Linear SDEs and Gaussian Posteriors
>
> The reviewer notes that linear SDE and Gaussian posterior assumptions may limit expressivity and whether these approximations affect performance.
>
> We acknowledge this in our paper (Section 3, Section 8). However, we note:
> 1. **These are essentially the same assumptions as made by all diffusion/flow models:**, and therefore we expect the performance to be strong and no more degraded than other models.
>
> 2. **These assumptions enable simulation-free training**, which is crucial for scalability and is a key advantage we share with observation-space diffusion models and SI.
>
> 3. **Empirical performance is strong**: Our ImageNet results (Table 1, Table 5) show competitive FID scores compared to observation-space models of similar size, demonstrating that these assumptions do not severely limit practical performance.

---

### Official Review · Reviewer_UcdG · 2025-11-02

**Soundness:** 4
**Presentation:** 4
**Contribution:** 4
**Rating:** 10
**Confidence:** 4

**Summary:**

This paper proposes Latent Stochastic Interpolants (LSI), which extend stochastic interpolants into a jointly trained latent-variable model by learning the encoder, decoder, and latent dynamics together through a continuous-time ELBO. The method gives a nice unifying view of SI inside latent models, keeps the flexibility of arbitrary priors, and preserves simulation-free training, while avoiding heavy computation in pixel space. I haven’t checked every technical detail, but the approach feels clean, scalable, and grounded in solid continuous-time modeling ideas. It seems to address a hard and meaningful problem in a practical way, and the flexibility to apply this direction beyond ImageNet is the kind of capability I’d like to see more of at ICLR.

**Strengths:**

* This paper tackles a non-trivial and meaningful problem in generative modeling.

* The approach of using joint training of encoder + latent dynamics + decoder feels principled and elegant.

* Flexible priors and continuous-time formulation are nice advantages.

* Experiments on ImageNet show competitive performance, and this general framework could extend to many domains.

* The paper is generally well written and easy to follow.

**Weaknesses:**

* Evaluation is mostly on ImageNet, so the broader impact still needs to be validated.

* It’s not fully clear how robust the training is across architectures and hyperparameters.

* The paper is heavily reliant on the supplementary materials and mathematical details.
This makes the paper less accessible to the readers who are not interested in going through the theoretical details.
The authors could provide the key insights in a more intuitive way, possibly using graphical illustrations.

**Questions:**

* How sensitive is performance to design choices in the latent SDE or encoder noise scale?

* Any results (even preliminary) on other modalities or conditional settings?

* Please provide some failure examples, and discuss the limitations of the proposed method.

---

> ### Author Response · Authors · 2025-11-19
> **Author Response**
>
> We are grateful for the reviewer's strongly positive assessment and for recognizing our work as "clean, scalable, and grounded in solid continuous-time modeling ideas". We appreciate the assessment that our approach is "principled and elegant" and addresses "a hard and meaningful problem in a practical way".
>
> ### Sensitivity to Design Choices
>
> We provide comprehensive ablations addressing this:
>
> **Encoder noise scale (Section 6, Figure 1 right)**: We systematically vary the encoder stochasticity parameter $c$ from 0 to beyond 0.06, showing:
> - Deterministic encoders ($c=0$) perform poorly (FID degrades significantly)
> - Performance improves with increasing $c$ until an optimal range
> - We also compare against learned variance, finding that fixed $c$ performs better in our experiments
>
> **Training objective parameterization (Section 4, Table 3)**: We evaluate four different parameterizations of the ELBO objective (OrigFlow, NoisePred, Denoising, InterpFlow), demonstrating that InterpFlow consistently achieves the best FID (3.76 vs. 4.56-4.73 for alternatives).
>
> **Loss trade-off weight $\beta$ (Section 6, Figure 1 left)**: We show how varying $\beta$ from $10^{-6}$ to $10^{-3}$ affects the balance between reconstruction (PSNR) and generation (FID), with FID improving from 4.53 to 3.75 before degrading when $\beta$ becomes too large.
>
> **Prior distribution $p_0$ (Table 4)**: We demonstrate robustness across diverse priors (Gaussian, Laplacian, Uniform, Gaussian Mixture), with all achieving competitive FID scores.
>
> These extensive ablations demonstrate that the method performs robustly across a range of settings.
>
> ### Results on Other Modalities or Conditional Settings
>
> We appreciate this suggestion for future work. Our paper focuses on establishing the **theoretical foundations** of Latent Stochastic Interpolants with rigorous mathematical derivations (Sections 2-5 and extensive appendices). To validate that our theory works in practice, we conduct comprehensive experiments on ImageNet, a challenging and standard benchmark for generative modeling.
>
> We do demonstrate conditional generation capability:
> - **All FID scores** are reported for **class-conditional** samples (Section 6)
> - **Figure 2 and Figure 7**: We show classifier-free guidance (CFG) works seamlessly with LSI, generating class-conditional samples with varying guidance weights $\lambda$
> - **Section 5**: We derive the guided drift formulation for conditional sampling
>
> Extending to other modalities (video, audio, etc.) would be valuable future work but is orthogonal to our core theoretical contribution.
>
> ### Failure Cases and Limitations
>
> **Theoretical limitations**: We discuss these in **Section 8 (Conclusion)**:
>
> > "However, to achieve scalable training, our approach makes simplifying assumptions for the variational posterior approximation. While restrictive, and common with other methods, these assumptions do not seem to limit the empirical performance".
>
> Specifically:
> - We assume **linear SDEs** with additive noise ($h_\phi(z_t,t) \equiv h_t z_t$, $\sigma(z_t,t) \equiv \sigma_t$) for tractable Gaussian transition densities (Section 3, Equation 6)
> - This enables **simulation-free sampling** of $z_t$, which is crucial for scalability
> - We acknowledge in Section 3: "Note that the assumptions made for eq. (6), while restrictive, do not limit the empirical performance".
>
> **Empirical failure modes**: Some of the typical failures, also common to diffusion and flow-based models, can be seen in **Figures 6 and 7** in appendix:
>
> - **Geometric distortions**: Objects can appear warped or have incorrect geometry (e.g., light switches in Figure 7, second-last row; measuring cup in Figure 7, fourth-last row)
> - **Text rendering**: Generated text is often garbled (e.g., Figure 7, third and sixth rows), a known challenge for pixel-space generative models
> - **Fine detail degradation**: Small-scale details can be inconsistent, particularly at lower guidance values ($\lambda = 0, 1$)
>
> These failure modes are **consistent with the broader literature** and tend to be more pronounced at lower classifier-free guidance weights, as shown in Figure 2 and Figure 7 where increasing $\lambda$ produces more coherent outputs.

---

### Official Review · Reviewer_TLxQ · 2025-11-03

**Soundness:** 3
**Presentation:** 3
**Contribution:** 3
**Rating:** 6
**Confidence:** 4

**Summary:**

The authors propose a latent stochastic interpolant method, where they aim to learn a generative model with a low-dimensional feature. To ensure the condition of the stochastic interpolant, they come up with a novel parameterization of the stochastic interpolant, conditional on the sample $z_0$ and the encoded output $z_1$. They further optimize the training objectives for reducing the variance in the training and improving the performance. The experiments were conducted on ImageNet and show noticeable improvements in the computational cost.

**Strengths:**

The goal of jointly optimizing the encoder and the generative model is challenging. And the authors' proposal seems to help address the problem.

The experiments on varying initial distributions show that the method might help to be used in the case where the Gaussian might not be ideal.

The overall presentation is clear.

**Weaknesses:**

The authors should consider giving a more detailed benchmark, including more models with a pre-trained encoder, and discuss the advantages over the comparison methods.

It would be good if the encoder is evaluated for its linear probing accuracy, since it's very useful to see if the encoder is meaningful or not.

**Questions:**

Could you show more results regarding the encoder's performance? How does it perform when you train the interpolant model using varying initial conditions and training objectives?

---

> ### Author Response · Authors · 2025-11-19
> **Author Response**
>
> We thank the reviwer for recognizing that our proposal addresses the challenging goal of jointly optimizing the encoder and generative model, and for acknowledging the clear presentation of our work.
>
>
> ### Detailed Benchmarks with Pre-trained Encoders
>
> We respectfully note that our primary contribution is **theoretical**: we derive a principled continuous-time ELBO that enables joint end-to-end learning of encoder, decoder, and latent generative models—something not previously possible with Stochastic Interpolants. While we rigorously prove our various results in appendix, the empirical studies serve to provide additional evidence in favor of our theoretical results.
> Comparison with a pre-trained encoder is significantly emphasized in our experimental studies ($\beta \rightarrow 0$), see Figure 1 (left panel), Table 2 and corresponding text in Section 6.
>
> Our approach specifically addresses the limitation that "arbitrary fixed latent representations may not be optimally aligned with the generative process" (Section 1). In fact, we empirically demonstrate in **Table 2** that joint training ($\beta > 0$) significantly outperforms independently trained models ($\beta \to 0$), with FID improving from 4.31 to 3.76 (≈13% improvement) even as we shift capacity away from the latent model. This validates that joint learning is beneficial—the central claim of our work.
> Our empirical studies are presented on the large scale ImageNet benchmark, using an
> architecture based on high performing prior art (see section N in appendix) and evaluated using the standard FID metric.
> The main goal of our experiments is to empirically support our theoretical results.
> A definite non-goal is to establish superiority of one architecture choice over another.
> While experimentation with other architectures would be interesting and may provide additional evidence, we believe that our rigorous proofs coupled with the current set of experimental studies already
> provide compelling evidence.
>
> ### Linear Probing Accuracy for Encoder
>
> We appreciate this suggestion.
> Evaluating the linear probing accuracy of encoder is quite interesting,
> though orthogonal to main contributions of the paper, namely enabling joint optimization of latent space stochastic interpolant model.
> We evalauted linear probing accuracy for 128x128 encoder and achieved $42 \\%$ ($\beta=1e-4$).
> Note that, from the point of view of variational approach to generative modeling, the encoder is simply a device that aids the training of the generative model, by approximating the posterior
> distribution of latents $p(z_1|x_1)$, and is "thrown away" once the model is trained, as it is not used during sampling.
> There is nothing in the objective that encourages linear separation of class labels.
> Further, **linear probing accuracy is not a standard evaluation metric for generative models**, as these models optimize for reconstruction and generation quality rather than discriminative features.
>
>
> **Our encoder evaluation**: Our encoder is explicitly optimized for generative modeling, not classification. We provide comprehensive evaluation through metrics appropriate for this objective:
>
> - **Section 6, Figure 1 (left)**: We extensively study the trade-off between reconstruction quality (PSNR) and generative performance (FID) by varying $\beta$, demonstrating that the encoder adapts to balance both objectives.
> - **Table 2**: Joint training maintains strong FID even when encoder capacity increases and latent model capacity decreases, showing the encoder learns effective representations for generation.
> - **Table 1**: Our encoder enables competitive FID scores while achieving 48-74% FLOP reduction during sampling.
>
> The appropriate metrics for evaluating our encoder are reconstruction quality (PSNR) and downstream generative quality (FID), both of which we report comprehensively and which demonstrate strong performance.
>
> ### Encoder Performance with Varying Conditions and Objectives
>
> We did not find any consistent patterns regarding reviewer suggested linear probing accuracy of the encoder across different $\beta$ values
> and accuracy remained low. We hypothesize that this is because the
> training objective optimizes the encoder for modeling the data distribution as opposed to linear separation of class labels.

---

### Author Response · Authors · 2025-11-19
**Shared author response**

We sincerely thank all reviewers for their thoughtful feedback and constructive comments. We are encouraged by the positive reception, particularly Reviewer UcdG's strong endorsement (Rating: 10) and recognition of our work as "a principled and elegant approach" that "addresses a hard and meaningful problem in a practical way". We appreciate that all reviewers acknowledged the soundness and good presentation of our work. We address the specific concerns raised by each reviewer in the corresponding responses.

---

### Author Response · Authors · 2025-11-26
**Gentle reminder**

Dear Reviewers,

We appreciate your valuable feedback and recognition. We have responded with a detailed rebuttal addressing your individual concerns. We hope that our responses helped address your concerns and are happy to answer any further questions.

Thanks,
-Authors

---

### Author Response · Authors · 2025-12-03
**Rebuttal/Discussion Summary for AC**

Dear AC,

We thank you and the reviewers for their time. Below is a summary of the consensus and the resolution of specific inquiries

**1. Unanimously positive reviews:** Initial reviews were **unanimously positive**, with ratings of one Strong Accept (10), and 2 Marginally above Accept (6) with no major concerns regarding the core contribution of joint latent space generative modeling formulation or its empirical verification.

**2. Strong Endorsement of Core Contribution (Reviewer UcdG, Rating: 10):**  strongly championed the work, describing the framework as a "principled and elegant" approach that offers a "unifying view" of stochastic interpolants in latent models. They specifically noted that the problem is "non-trivial and meaningful" and recommended the paper be highlighted (spotlight/oral, Rating 10). We have responsed to additional clarifications in our detailed response.

**3. Resolution of Empirical Questions (Reviewer TLxQ, Rating 6):** Reviewer found the proposal "clear". Their main critique focused on benchmarking the encoder (specifically via linear probing). In our rebuttal, we provided a detailed response distinguishing our goal (joint latent space generative modeling) from pure representation learning benchmarks. Notably, from the generative modeling perspective, encoder is a “throwaway” training aid that is never used at test time for sampling from model. We also provided a point evaluation of the encoder. We believe we have addressed the concerns regarding the scope of evaluation in our detailed response to the reviewer.

**4. Clarification on Theoretical Scope (Reviewer xrNR, Rating: 6):** Reviewer found the derivation "theoretically grounded" and the paper "well motivated". Their primary comment regarding the expressivity of Gaussian posteriors was a point we have already discussed as a limitation. Their follow-up regarding "learnable priors" was a query about future extensions rather than a critique of the current work. In our detailed response we clarified that our framework supports arbitrary priors, including learnable ones.

**Conclusion:** All reviewers agree on the **soundness of the continuous-time ELBO derivation** and the **effectiveness of the joint training scheme**. With a Strong Accept rating and no remaining major concerns regarding the method's validity or novelty, we hope the AC agrees the paper is ready for acceptance.

Best regards,
Authors

---

### Meta-Review · Area_Chair_wBzG · 2026-01-06

**Summary:**

The manuscript introduces Latent Stochastic Interpolants, which generalize Stochastic Interpolants to enable joint end-to-end training of an encoder, a decoder, and a generative model operating entirely within the learned latent space. The idea of the approach is natural and it is validated through numerical experiments. Overall this is a solid contribution.

**Reviewer Concerns:**

The authors have addressed all reviewer's concerns.

**Reviewer Scores:**

I believe the reviewers will keep their scores.

---

### Decision · Program_Chairs · 2026-01-26

Accept (Poster)